# Hemispheric asymmetry of tau pathology is related to asymmetric amyloid deposition in Alzheimer's Disease

Toomas Erik Anijärv [1] ✉, Rik Ossenkoppele [1,2,3], Ruben Smith [1,4], Alexa Pichet Binette [1,5,6], Lyduine E. Collij [1,7,8], Harry H. Behjat [1], Jonathan Rittmo [9], Linda Karlsson [1], Khazar Ahmadi [10], Olof Strandberg[1], the Alzheimer's Disease Neuroimaging Initiative*, Danielle van Westen [11], Jacob W. Vogel [9], Erik Stomrud[1,4], Sebastian Palmqvist [1,4], Niklas Mattsson-Carlgren [1,4,12], Nicola Spotorno [1,25] ✉ & Oskar Hansson [1,25] ✉

The distribution of tau pathology in Alzheimer's disease (AD) shows remarkable inter-individual heterogeneity, including hemispheric asymmetry. However, the factors driving this asymmetry remain poorly understood. Here we explore whether tau asymmetry is linked to i) reduced inter-hemispheric brain connectivity (potentially restricting tau spread), or ii) asymmetry in amyloid-beta (Aβ) distribution (indicating greater hemisphere-specific vulnerability to AD pathology). We include 452 participants from the Swedish BioFINDER-2 cohort with evidence of both Aβ pathology (CSF Aβ42/40 or neocortical Aβ-PET) and tau pathology (temporal tau-PET), categorising them as left asymmetric (n = 102), symmetric (n = 306), or right asymmetric (n = 44) based on temporal lobe tau-PET uptake distribution. We assess edge-wise inter-hemispheric functional (RSfMRI; n = 318) and structural connectivity (dMRI; n = 352) but find no association between tau asymmetry and connectivity. In contrast, we observe a strong association between tau and Aβ laterality patterns based on PET uptake (n = 233; β = 0.632, p < 0.001), which we replicate in three independent cohorts (n = 234; β = 0.535, p < 0.001). In a longitudinal Aβ-positive sample, we show that baseline Aβ asymmetry predicts progression of tau laterality over time (n = 289; β = 0.025, p = 0.028). These findings suggest that tau asymmetry is not associated with a weaker inter-hemispheric connectivity but might reflect hemispheric differences in vulnerability to Aβ pathology, underscoring the role of regional vulnerability in determining the distribution of AD pathology.

Alzheimer's disease (AD), the leading cause of dementia worldwide, exhibits substantial heterogeneity in its pathological manifestations and clinical progression[1,2]. The disease is characterised by two primary pathological hallmarks: the accumulation of amyloid-beta (Aβ) plaques throughout the neocortex and the progressive aggregation of hyperphosphorylated tau proteins forming neurofibrillary tangles, ultimately resulting in neurodegeneration and cognitive decline[3]. The accumulation of tau pathology in AD is commonly described as

A full list of affiliations appears at the end of the paper. *A list of authors and their affiliations appears at the end of the paper.
✉e-mail: toomas_erik.anijarv@med.lu.se; nicola.spotorno@med.lu.se; oskar.hansson@med.lu.se

following a stereotypical distribution[4,5]. Nonetheless, tau-PET studies have demonstrated heterogeneity in the distribution of tau across individuals and multiple spatiotemporal patterns have been described, differentially associated with cognitive functioning and decline[1,2,6–8].

One of the manifestations of this heterogeneity is an asymmetric distribution of tau pathology between the two hemispheres. Hemispheric asymmetry of tau pathology in AD has been associated with younger age, more severe pathological burden and rapid multi-domain cognitive impairment[7,9–11]. Moreover, some studies indicate that tau asymmetry is more easily identified during the more advanced stages of the disease[7,12], often with left hemisphere dominance[13]. However, asymmetrical tau distribution has also been found to be present in preclinical AD[10]. More commonly, asymmetry in tau distribution is presented in atypical AD cases, such as posterior cortical atrophy (PCA)[14] and primary progressive aphasia (PPA)[15] where PCA can present with both left and right predominant tau distribution while PPA is usually characterized by left-sided asymmetry[16–18]. Furthermore, a data-driven approach to tau-PET data in a large multi-cohort study[7], uncovered four primary AD subtypes characterized by different spatiotemporal profiles of tau pathology. Among these, the lateral temporal subtype, comprising 19% of cases, exhibited strong tau asymmetry, while the remaining subtypes demonstrated more moderate patterns of tau lateralisation.

Although previous work indicates that asymmetric distribution of tau pathology is relatively common, the underlying drivers of this phenomenon are not clearly characterised. Two primary mechanisms have been proposed to explain tau accumulation patterns. The first suggests that misfolded tau proteins spread through connected brain regions in a prion-like manner[19–23]. The second mechanism proposes that tau accumulation is primarily driven by local replication or regional vulnerability, which can be influenced by various factors including the local presence of Aβ pathology[24–26]. These mechanisms are not mutually exclusive. Understanding their contribution to hemispheric tau asymmetry could reveal critical insights into AD pathophysiology and guide the development of novel therapeutic strategies. In this study, we proposed two possible hypotheses for explaining asymmetric tau distribution. First, we investigated whether individuals characterized by asymmetric tau patterns display reduced inter-hemispheric functional and structural connectivity, possibly indicating reduced tau spreading between hemispheres. Second, we explored the idea that this asymmetry in tau accumulation is related to hemispheric differences in Aβ deposition, suggesting that these individuals display greater regional vulnerability to AD pathology in one hemisphere more than the other.

## Results

### Participants

A sample of 837 Aβ-positive (A+) participants, based on neocortical Aβ-PET or CSF Aβ42/40, with available tau-PET scan(s) from the BioFINDER-2 cohort was included in this study. The cohort consisted of both clinically unimpaired and clinically impaired participants (see 'Methods – Main cohort' for detailed information). A cross-sectional sample of subjects with evidence of tau pathology (A+T+; n = 475) based on unilateral (i.e., most affected hemisphere) tau-PET uptake in the temporal meta region of interest (meta-ROI; SUVR > 1.362[27,28]; regions detailed in Table S1.1) was selected and categorized into three groups according to the spatial distribution of tau. Specifically, a tau laterality index (LI) was computed for each participant based on the tau-PET uptake in a temporal meta-ROI covering the regions described by Braak stages I-IV. For the following analyses, tau laterality was examined both as a continuous measure (tau LI) and as a grouping factor. Three groups were defined: participants with LI exceeding ±1 SD from perfect symmetry (i.e., LI = 0) were assigned to the right asymmetric (RA; n = 44) or left asymmetric (LA; n = 102) group, respectively, while those within ±1 SD were assigned to the symmetric

(S; n = 306) group. Subjects with a borderline laterality index, defined as a ±5% interval around the threshold were not allocated to any group and excluded from the analyses (n = 23), resulting in a final sample size of 452 (Table 1; Fig. 1a,b).

Demographic characteristics, including age, sex, *APOE* ε4 allele carriership, and cognitive performance did not differ between the three groups. Moreover, average bilateral global (i.e., both whole hemispheres) tau load did not differ between the three groups, nor did Aβ (Table 1; Fig. 1c). SUVR values from the temporal meta-ROI showed a trend toward higher average bilateral tau load in participants with asymmetric tau distribution compared to those with symmetric distribution, though this difference was statistically significant only between the right tau asymmetric group and the symmetric group (t = 2.688, p = 0.023). When examining unilateral tau load in the temporal meta-ROI in the most affected hemisphere, both asymmetric groups showed significantly higher load compared to the symmetric group (LA-S: t = 4.828, p < 0.001; RA-S: t = 5.240, p < 0.001). To account for these differences, SUVR values from average bilateral global tau load were included as a covariate in subsequent analyses. The A+ T+ sample included 15 cases with atypical presentation of AD. Among those with left-sided tau asymmetry, four individuals were diagnosed with PPA and one with PCA. Within the group characterized by symmetric tau deposition, three cases were diagnosed with PPA and seven

**Table 1 | Demographics**

| | | Cross-sectional A+T+ (n = 452) | | | |
|---|---|---|---|---|---|
| | | LA (n = 102; 22%) | S (n = 306; 68%) | RA (n = 44; 10%) | *P*-value |
| Age, years | | 73.4 (6.7) | 73.5 (7.3) | 72.1 (6.9) | 0.462 |
| Sex | M | 46 (45%) | 135 (44%) | 18 (41%) | 0.895 |
| | F | 56 (55%) | 171 (56%) | 26 (59%) | |
| Education, years[a] | | 12.7 (3.82) | 12.5 (4.0) | 13.4 (4.1) | 0.354 |
| Handedness[b] | R | 85 (94%) | 262 (94%) | 41 (100%) | 0.567 |
| | L | 5 (6%) | 16 (6%) | | |
| | A | | 1 (<1%) | | |
| Diagnosis | CU | 11 (11%) | 40 (13%) | 9 (21%) | 0.511 |
| | MCI | 39 (38%) | 106 (35%) | 12 (27%) | |
| | AD | 52 (51%) | 160 (52%) | 23 (52%) | |
| Braak stage | I-II | | 11 (4%) | | <0.001 |
| | III-IV | 21 (20%) | 107 (35%) | 6 (14%) | |
| | V-VI | 81 (80%) | 188 (61%) | 38 (86%) | |
| Temporal tau, SUVR | | 1.9 [1.7, 2.3] | 1.7 [1.4, 2.3] | 2.1 [1.8, 2.7] | <0.001 |
| Temporal tau LI | | −14.4 (3.7) | −0.4 (4.9) | 14.7 (5.4) | <0.001 |
| Neocortical Aβ, SUVR[c] | | 1.59 (0.20) | 1.63 (0.23) | 1.56 (0.23) | 0.211 |
| *APOE* ε4 | 0 | 32 (31%) | 75 (25%) | 16 (36%) | 0.188 |
| | 1 | 49 (48%) | 182 (59%) | 21 (48%) | |
| | 2 | 21 (21%) | 49 (16%) | 7 (16%) | |
| MMSE[d] | | 24.0 [21.0, 27.0] | 25.0 [21.0, 27.0] | 26.0 [22.7, 28.0] | 0.328 |
| mPACC[e] | | −2.7 [−4.0, −1.7] | −2.6 [−3.8, −1.5] | −2.5 [−3.9, −1.2] | 0.568 |

Categorical variables have been presented as 'count (%)', normally distributed continuous variables as 'mean (SD)' and non-normally distributed variables as 'median [interquartile range]'. All variables were compared between the tau asymmetry groups using either one-way ANOVA, Kurskal-Wallis, or Chi-squared test depending on the type and distribution of the data. *LA* left tau asymmetric, *S* tau symmetric, *RA* right tau asymmetric, *M* male, *F* female, *R* right-handed, *L* left-handed, *A* ambidextrous, *CU* cognitively unimpaired, *MCI* mild cognitive impairment, *AD* Alzheimer's disease, *SUVR* standardized uptake value ratio, *LI* laterality index, *Aβ* amyloid-beta, *MMSE* Mini-Mental State Examination, *mPACC* modified Preclinical Alzheimer Cognitive Composite. Missing data for 10[a], 42[b], 219[c], 1[d], and 42[e] individuals.

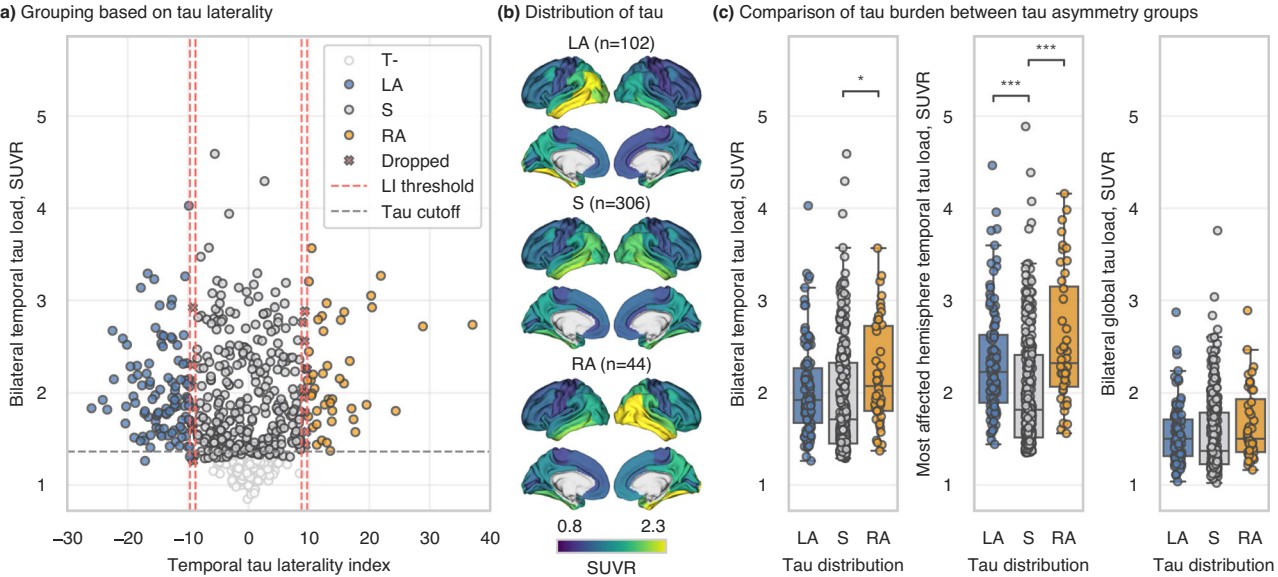

**(a)** Grouping based on tau laterality

**(b)** Distribution of tau

**(c)** Comparison of tau burden between tau asymmetry groups

**Fig. 1 | Grouping of the subjects. a** Participants were divided into three groups based on the distribution of temporal tau-PET uptake; **b** Average tau-PET SUVRs for each group; **c** Comparison of tau load between the groups. Boxplots in panel **c** represent tau uptake across the three tau asymmetry groups. The horizontal line within each box indicates the median, while the lower and upper box edges denote the first and third quartiles, respectively. Whiskers extend to 1.5 times the inter-quartile range, and dots represent individual data points. Statistical comparisons between the groups in panel **c** were performed using ordinary least squares

multiple linear regression (tau load - age + sex + group). Visualised statistical annotations indicate significance levels Bonferroni-corrected for the number of group comparisons. The sample sizes for each group compared in panel **c** are following: $n = 102$ for LA, $n = 306$ for S, and $n = 44$ for RA. Laterality index (%) = 100 × (right tau − left tau) / (right tau + left tau). LA left tau asymmetric, S tau symmetric, RA right tau asymmetric, SUVR standardized uptake value ratio; *, $p_{Bonf} < 0.05$; ***, $p_{Bonf} < 0.001$.

with PCA. No atypical AD cases were identified in the right tau asymmetric group.

**Inter-hemispheric connectivity does not differ between participants with asymmetric and symmetric pattern of tau deposition**
Subsequently, we assessed whether asymmetric tau distribution is associated with differences in inter-hemispheric brain connectivity. Connectivity was calculated between Desikan-Killiany atlas regions using functional connectivity (correlation of blood-oxygen-level-dependent signals, BOLD) and structural connectivity (anatomically-constrained tractography). To ensure biological relevance, only the top 10% of inter-hemispheric connections identified in a separate sample of healthy controls were retained for analysis (see 'Methods' for detailed information).

No statistically significant differences were found in average inter-hemispheric functional connectivity ($n = 318$; S-LA: $β = −0.151$, 95%CI = [−0.417; 0.116], $p_{Bonf} = 0.802$; S-RA: $β = 0.076$, 95%CI = [−0.312; 0.463], $p_{Bonf} > 0.9$) or structural connectivity ($n = 352$; S-LA: $β = 0.135$, 95% CI = [−0.107; 0.376], $p_{Bonf} = 0.823$; S-RA: $β = 0.170$, 95%CI = [−0.162; 0.502], $p_{Bonf} > 0.9$) between the tau asymmetric groups and the tau symmetric group (Fig. 2c). Similarly, no associations were found between absolute global tau laterality index and average inter-hemispheric functional connectivity ($β = 0.090$, 95%CI = [−0.027; 0.207], $p = 0.130$) or structural connectivity ($β = −0.037$, 95%CI = [−0.144; 0.071], $p = 0.502$) when all A+T+ individuals were assessed (Fig. 2a). Notably, edge-wise analyses of homotopic (i.e., inter-hemispheric same-region) connectivity ($n = 36$ connections) revealed no significant associations between absolute tau laterality and functional (all $p_{FDR} > 0.6$) or structural connectivity (all $p_{FDR} > 0.6$) (see Supplementary S2). Moreover, performing a whole-brain connectome analysis (i.e., not limited to only inter-hemispheric connections) using Network Based Statistics (NBS) did not show evidence of reduced functional or structural connectivity between hemispheres when

comparing the asymmetric and symmetric groups (see Fig. S2.3 and Table S2.1 in Supplementary S2).

We further investigated microstructural integrity within the main white matter tracts connecting the two hemispheres but found no statistically significant associations between absolute global tau laterality and fractional anisotropy (Fig. 2b) - in the corpus callosum ($β = −0.047$, 95%CI = [−0.163, 0.069], $p = 0.424$), forceps major ($β = 0.022$, 95%CI = [−0.093, 0.137], $p = 0.708$), or forceps minor ($β = −0.021$, 95%CI = [−0.137, 0.095], $p = 0.721$). This result was consistent with the comparison of the same measure between the groups defined based on the tau laterality index (Fig. 2d) - in the corpus callosum (S-LA: $β = 0.042$, 95%CI = [−0.216; 0.300], $p_{Bonf} > 0.9$; S-RA: $β = −0.134$, 95%CI = [−0.518; 0.251], $p_{Bonf} > 0.9$), forceps major (S-LA: $β = −0.033$, 95%CI = [−0.289; 0.223], $p_{Bonf} > 0.9$; S-RA: $β = −0.220$, 95% CI = [−0.599; 0.159], $p_{Bonf} = 0.762$), or forceps minor (S-LA: $β = 0.001$, 95%CI = [−0.258; 0.260], $p_{Bonf} > 0.9$; S-RA: $β = −0.136$, 95%CI = [−0.520; 0.248], $p_{Bonf} > 0.9$); similar results were found for mean diffusivity (see Fig. S2.1 in Supplementary S2).

**Sensitivity analyses.** While our primary analyses found no association between average brain connectivity and tau laterality, we further investigated whether connectivity relates to overall tau burden (see Supplementary S2). Compared to A-T- cognitively unimpaired controls, A+T+ individuals exhibited significantly lower average connectivity across the whole brain (functional: $β = −0.184$, $p = 0.028$; structural: $β = −0.387$, $p < 0.001$) including inter-hemispheric connections (functional: $β = −0.186$, $p = 0.027$; structural: $β = −0.190$, $p = 0.012$) after adjusting for age and sex (Fig. S2.4). Edge-wise, within A+T+ individuals, lower functional and structural connectivity between homotopic regions was associated with higher bilateral tau burden in multiple regions across the brain, with the strongest effects observed in occipital and temporal areas for functional connectivity and across the neocortex except the temporal lobe for structural

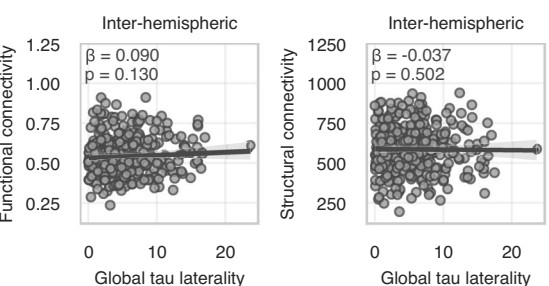

**(a)** Total inter-hemispheric connectivity vs absolute global tau laterality

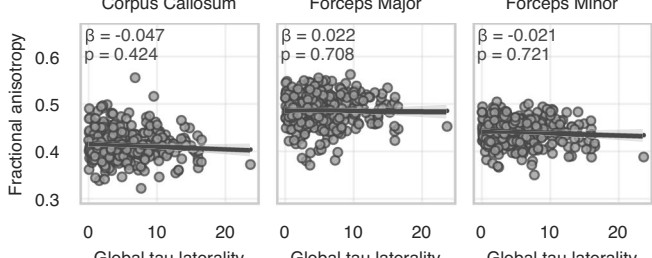

**(b)** White matter microstructural integrity vs absolute global tau laterality

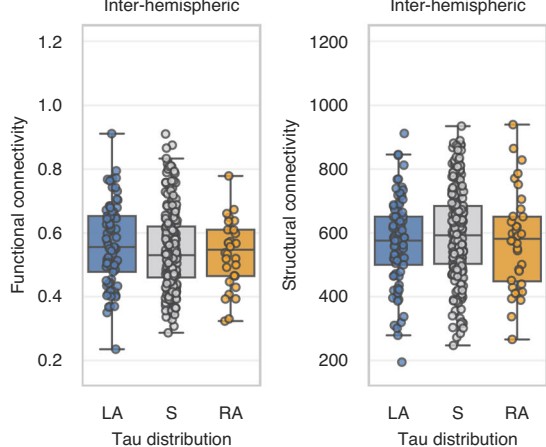

**(c)** Total inter-hemispheric connectivity between tau asymmetry groups

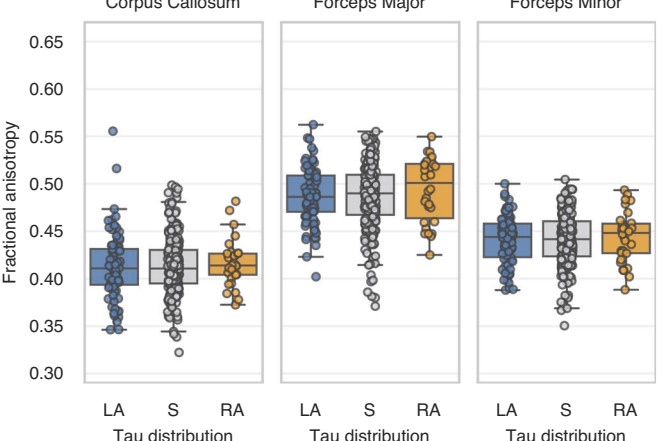

**(d)** White matter microstructural integrity between tau asymmetry groups

**Fig. 2 | Association between average connectivity between hemispheres and asymmetry in tau distribution. a** Inter-hemispheric functional/structural connectivity vs absolute global tau laterality; **b** Fractional anisotropy in main white matter tracts vs absolute global tau laterality; **c** Inter-hemispheric functional/structural connectivity between tau asymmetry groups; **d** Fractional anisotropy in main white matter tracts between tau asymmetry groups. Panels **a** and **b** show regression lines with 95% confidence intervals, with statistical annotations indicating the standardized effect size and significance level of tau laterality as a predictor of connectivity in ordinary least squares multiple linear regression models (inter-hemispheric connectivity ~ age + sex + global tau load + global tau laterality). Boxplots in panels **c** and **d** represent inter-hemispheric connectivity across the

three tau asymmetry groups, where the groups were statistically compared using ordinary least squares multiple linear regression models (inter-hemispheric connectivity ~ age + sex + global tau load + group), with the significance levels Bonferroni-corrected for the number of group comparisons. The horizontal line within each box indicates the median, while the lower and upper box edges denote the first and third quartiles, respectively. Whiskers extend to 1.5 times the inter-quartile range, and dots represent individual data points. Note: 16 subjects were dropped from the analyses within panels **b** and **d** after visual quality control of the tract segmentation, resulting in $n = 336$. LA left tau asymmetric, S tau symmetric, RA right tau asymmetric.

connectivity (Fig. S2.2). Second, to address potential confounding effects of regional variability in tau load, three additional composite ROIs were defined based on the ranking of group-average tau SUVR values from A+T+ subjects (i.e., high tau burden corresponds to regions above the 75th percentile, medium tau burden corresponds to regions between the 25th and the 75th percentile and low tau burden corresponds to regions below the 25th percentile; see Supplementary S2). After adjusting for age, sex, and bilateral tau load, neither average inter-hemispheric (Fig. S2.5; functional: all $p_{Bonf} > 0.8$; structural: all $p_{Bonf} > 0.2$) nor intra-hemispheric connectivity (Fig. S2.6; functional: all $p_{Bonf} > 0.4$; structural: all $p_{Bonf} > 0.5$) showed significant associations with absolute tau laterality in any of these composite ROIs. Together, these sensitivity analyses confirm the null results we observed in the primary connectivity analyses.

### Strong associations between the laterality of Aβ and tau pathologies

We then assessed whether tau asymmetry is related to hemispheric differences in vulnerability to Aβ pathology. Within the A+T+ sample with available Aβ-PET (n = 233), there was a strong association between the degree of global laterality in tau and in Aβ pathologies (Fig. 3a; $β = 0.632$, 95%CI = [0.530; 0.733], $p < 0.001$); however, the magnitude of asymmetry was greater in tau than in Aβ pathology. Additional

analyses based on a priori selected meta-ROIs revealed the strongest associations between global Aβ laterality and tau laterality were present in a meta-ROI encompassing regions included in Braak stages III-IV (Fig. S2.7; $β = 0.655$, 95%CI = [0.556; 0.754], $p_{Bonf} < 0.001$). Similar results were found when investigating the associations at a regional level with all regions (min $β = 0.167$; mean $β = 0.380$; max $β = 0.633$; all $p_{FDR} < 0.05$) except for two regions (the rostral anterior cingulate [$β = 0.130$, $p_{FDR} = 0.051$] and the hippocampus [$β = -0.034$, $p_{FDR} = 0.608$]). The strongest effect sizes were found in the temporal regions, particularly the inferior temporal ($β = 0.633$, $p_{FDR} < 0.001$), fusiform ($β = 0.613$, $p_{FDR} < 0.001$), and middle temporal ($β = 0.571$, $p_{FDR} < 0.001$) gyri (Fig. 3b). Moreover, tau laterality appeared to be related to Aβ laterality not only in the same regions but across the brain. In particular, temporal regions showed the strongest average association with all brain regions (inferior temporal: $β_{avg} = 0.428$; fusiform: $β_{avg} = 0.405$; middle temporal: $β_{avg} = 0.405$; see Fig. S2.8 in Supplementary S2).

**Replication in independent cohorts.** To confirm the association between the laterality of Aβ and tau, we performed the same analyses on three independent cohorts – Open Access Series of Imaging Studies (OASIS-3)[29], Anti-Amyloid Treatment in Asymptomatic Alzheimer's Disease (A4)[30,31], and Alzheimer's Disease Neuroimaging Initiative

**(a)** Average hemispheric tau laterality vs Aβ laterality          **(b)** Region-by-region tau laterality vs Aβ laterality

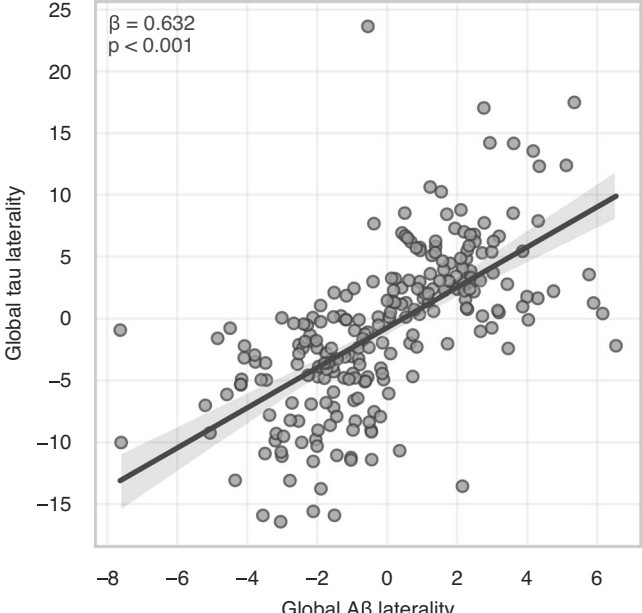
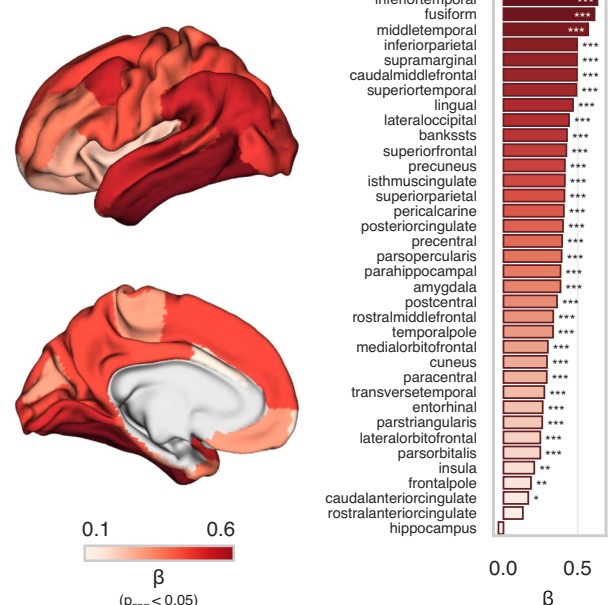

**Fig. 3 | Association between asymmetry in Aβ and tau distribution. a** Aβ and tau laterality averaging the PET uptake over the whole hemisphere; **b** Regional-specific asymmetries in Aβ and tau. Panel **a** shows a regression line with 95% confidence interval, with statistical annotation indicating the standardized effect size and significance of Aβ laterality as a predictor of tau laterality in an ordinary least squares multiple linear regression model (global tau laterality ~ age + sex + global Aβ laterality). Panel **b** depicts similar models for all homotopic regions, with significance levels FDR-corrected and visualized on brain surface and bar plots. Aβ amyloid-beta; *, $p_{FDR} < 0.05$; **, $p_{FDR} < 0.01$; ***, $p_{FDR} < 0.001$.

**(a)** Combined external cohorts          **(b)** All external cohorts separately

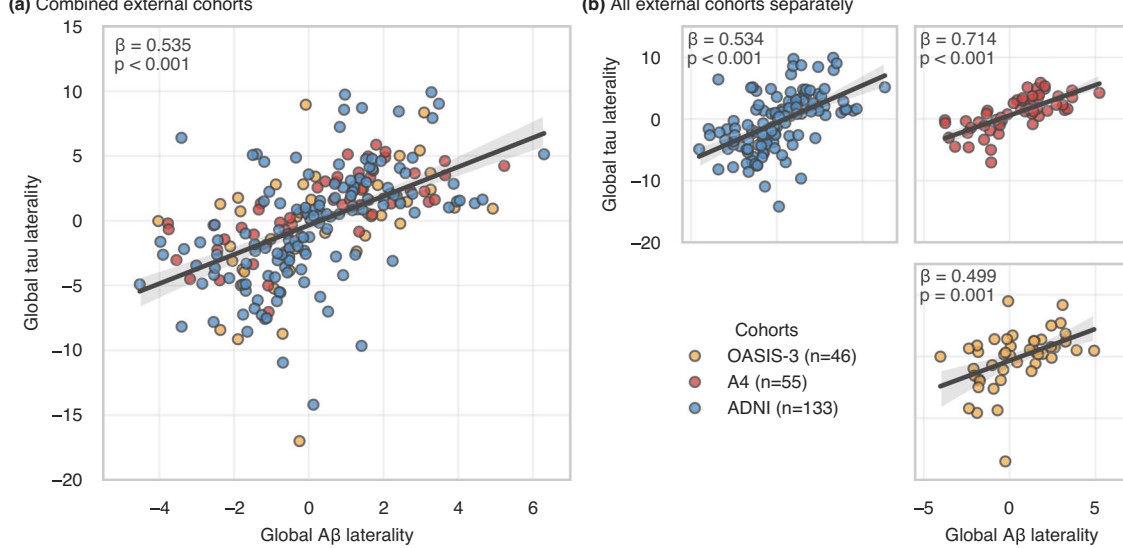

**Fig. 4 | Association between global Aβ laterality and tau laterality in external cohorts. a** All cohorts combined; **b** Cohorts separately. Both panels show regression lines with 95% confidence intervals, with statistical annotations indicating the standardized effect size and significance of Aβ laterality as a predictor of tau laterality in ordinary least squares multiple linear regression models (global tau laterality ~ age + sex + cognitive impairment status + global Aβ laterality). Aβ amyloid-beta, OASIS-3 Open Access Series of Imaging Studies, A4 Anti-Amyloid Treatment in Asymptomatic Alzheimer's Disease, ADNI Alzheimer's Disease Neuroimaging Initiative.

(ADNI)[32] – out of which only A+T+ subjects were included (see Supplementary Table S2.9 for demographics). We first combined the three cohorts in a single analysis controlling for the heterogeneity in the disease stages represented in the different cohorts (i.e., cognitively unimpaired or impaired). The strong relationship between asymmetries of global Aβ and tau distributions was replicated with a similar effect size compared to our main cohort (Fig. 4a; β = 0.535, 95%CI = [0.425; 0.645], p < 0.001). When analysed separately, each unique cohort displayed similar associations between Aβ and tau laterality despite the smaller sample sizes (Fig. 4b) – A4 (n = 55; β = 0.714, 95% CI = [0.516; 0.913], p < 0.001), OASIS-3 (n = 46; β = 0.499, 95%CI = [0.208; 0.789], p = 0.001), ADNI (n = 133; β = 0.534, 95%CI = [0.386;

0.683], p < 0.001). Interestingly, all replication cohorts showed the greatest effect size of spatial relationship between the laterality of tau and Aβ pathologies in regions within Braak stages III and IV (see Fig. S2.9 in Supplementary 2).

**Sensitivity analyses.** To support that the results did not arise from methodological issues, we performed a set of sensitivity analyses in the main cohort (see Supplementary S2 for a detailed overview). First, we investigated the association between the two pathologies using PET SUVR values corrected for partial volume effects (see Fig. S2.10) and found consistent results across the brain (global: β = 0.648, p < 0.001) and within all meta-ROIs defined based on Braak stages (I-II: β = 0.178, $p_{Bonf}$ = 0.016; III-IV: β = 0.728, $p_{Bonf}$ < 0.001; V-VI: β = 0.590, $p_{Bonf}$ < 0.001). Second, the association between the laterality of tau and Aβ remained essentially the same when adjusting for laterality of cerebral blood flow measured with arterial spin labelling (ASL; global: β = 0.498, p < 0.001) or cortical thickness (global: β = 0.560, p < 0.001) in the model (see Fig. S2.12). To explore the estimated time differences in Aβ pathology onset between hemispheres in tau asymmetry groups, we applied the Sampled Iterative Local Approximation (SILA) algorithm on the full BioFINDER-2 cohort. SILA models individual Aβ accumulation curves to estimate the timing of hemispheric Aβ onset (see Supplementary S2). Asymmetric tau groups exhibited significantly larger inter-hemispheric differences in estimated global Aβ onset (left asymmetric: Δ = 1.7 years, $p_{Bonf}$ = 0.006; right asymmetric: Δ = 2.5 years, $p_{Bonf}$ < 0.001) compared to the symmetric group (Δ = 1.3 years; Fig. S2.13), suggesting earlier regional Aβ accumulation may contribute to tau lateralisation. Finally, Aβ-PET and tau-PET scans for nine representative cases (i.e., three cases for each asymmetrical profile) can be seen in Supplementary S3.

### Higher baseline Aβ laterality is associated with increased tau laterality over time

To further investigate the impact of baseline Aβ laterality on longitudinal change in tau laterality, we examined a subsample of 289 A+ subjects from the original cohort of 837 who underwent at least one Aβ-PET scan and additionally had available longitudinal tau-PET scans (range of timepoints = 2–5; average follow-up = 2.9 years). Within all these A+ subjects, a linear mixed effect (LME) model with only time as a predictor displayed an increasing tau asymmetry longitudinally (global: β = 0.043, 95%CI = [0.016; 0.069], p = 0.002). Next, to investigate the effects Aβ laterality on tau laterality, we included baseline Aβ laterality and covariates to the LME model. Within the same sample, higher baseline Aβ laterality was predictive of changes over time in tau laterality (Fig. 5a; global: β = 0.025, 95%CI = [0.003; 0.048], p = 0.028), i.e., greater asymmetry of the Aβ distribution at baseline was associated with greater asymmetry of the tau distribution over time.

We then further stratified the sample based on baseline temporal meta-ROI tau-PET uptake to A+T- (n = 180) and A+T+ (n = 109) groups (see Table S2.2 for demographics) and repeated the analyses according to regions defined in Braak stages. In the A+T- group (Fig. 5b), the association between Aβ laterality and tau laterality over time was confirmed in Braak stages I-II (β = 0.054, 95%CI = [0.017; 0.091], $p_{Bonf}$ = 0.012) and III-IV (β = 0.078, 95%CI = [0.038; 0.119], $p_{Bonf}$ < 0.001). In contrast, Aβ laterality was not associated with changes over time in tau laterality in the A+T+ group (Fig. 5c). For an overview of the models, see Supplementary Table S2.3, S2.4, and S2.5.

To further assess how pathological progression affects the interaction between baseline Aβ laterality and tau laterality over time, we stratified the A+T- group into additional two subgroups – individuals who remained A+T- throughout their follow-up (n = 142; average follow-up = 2.9 years) and those who progressed to A+T+ during follow-up (n = 38; average follow-up = 3.5 years). Subjects who maintained A+T- status (Fig. 6a) showed a significant association between baseline Aβ laterality and tau laterality over time in Braak I-II (β = 0.076, 95%

CI = [0.033; 0.119], $p_{Bonf}$ = 0.002), but not in Braak III-IV or V-VI. Conversely, subjects who progressed to A+T+ (Fig. 6b) exhibited strong associations between baseline Aβ laterality and tau laterality over time in Braak III-IV (β = 0.131, 95%CI = [0.075; 0.188], $p_{Bonf}$ < 0.001) and V-VI (β = 0.144, 95%CI = [0.060; 0.227], $p_{Bonf}$ = 0.002), but not in Braak I-II. For an overview of the models, see Supplementary Table S2.6 and S2.7.

**Sensitivity analyses.** To further confirm that these findings were not a result of our methodological choices, we repeated the analysis using partial volume corrected PET SUVR values (see Supplementary S2). Most of the findings were successfully replicated with most notably the full A+ sample exhibiting even stronger effect (see Fig. S2.11; β = 0.041, 95%CI = [0.021; 0.060], p < 0.001) than our main analysis, but there were also some exceptions. For instance, in the sensitivity analysis the A+T+ individuals displayed a significant interaction effect in Braak V-VI (β = 0.064, 95%CI = [0.034; 0.094], $p_{Bonf}$ < 0.001), which we did not detect in the main analysis. Moreover, individuals who converted from A+T- to A+T+ showed an interaction effect of baseline Aβ laterality on tau laterality over time, only trending towards significance at Braak III-IV (β = 0.081, 95%CI = [0.015; 0.147], $p_{Bonf}$ = 0.051) and V-VI (β = 0.093, 95%CI = [0.014; 0.171], $p_{Bonf}$ = 0.061). Additionally, we tested whether baseline tau or Aβ asymmetry predicted faster tau accumulation. Higher baseline absolute tau laterality was associated with a steeper increase in tau load in Braak III-IV (β = 0.085, $p_{Bonf}$ < 0.001) and V-VI (β = 0.114, $p_{Bonf}$ < 0.001) regions, while baseline Aβ laterality showed no effect (see Fig. S2.14). Finally, we evaluated whether region-specific tau laterality was influenced by inter-hemispheric connectivity alongside Aβ asymmetry. Neither functional nor structural connectivity improved model fits (all $p_{FDR}$ > 0.6) or reached significance as predictors, indicating negligible explanatory power beyond Aβ asymmetry (see Supplementary S2).

### Tau asymmetry in relation to cognitive decline

A+ participants showed distinct patterns of cognitive decline based on the degree of tau asymmetry (i.e., absolute laterality index). Higher baseline tau laterality was associated with a steeper decline in modified Preclinical Alzheimer Cognitive Composite (mPACC) scores (Fig. 7a; n = 259) in Braak III-IV (β = −0.132, 95%CI = [−0.171; −0.094], $p_{Bonf}$ < 0.001) and V-VI (β = −0.157, 95%CI = [−0.194; −0.121], $p_{Bonf}$ < 0.001). However, after adjusting for average tau uptake at the corresponding meta-ROIs, the independent effect of tau laterality on mPACC over time was statistically significant only in regions corresponding to Braak V-VI (Fig. 7b; β = −0.104, 95%CI = [−0.153; −0.055], $p_{Bonf}$ < 0.001). Direction of tau lateralisation (left vs right asymmetry) did not influence cognitive trajectories (see Fig. S2.15 in Supplementary S2). Aβ laterality did not have any significant effect on mPACC scores (Fig. 7c). See Supplementary Table S2.8 for an overview of the models.

## Discussion

In this work, we investigated potential mechanisms underlying hemispheric asymmetry in tau distribution in AD. In contrast to one of our original hypotheses proposing reduced connectivity between hemispheres as one of the main contributing factors for asymmetrical distribution of tau, we found no evidence of an alteration in inter-hemispheric connectivity in participants with asymmetric distribution of tau compared to participants with a symmetric distribution of tau using neither functional nor structural connectivity. Moreover, the main white matter tracts connecting the two hemispheres showed similar levels of microstructural integrity assessed by fractional anisotropy and mean diffusivity. These null findings suggest that inter-hemispheric connectivity differences do not account for the asymmetric distribution of tau in individuals along the AD continuum. However, connectivity-based spreading of tau may still occur during the very early stages of the disease, before the pathology becomes

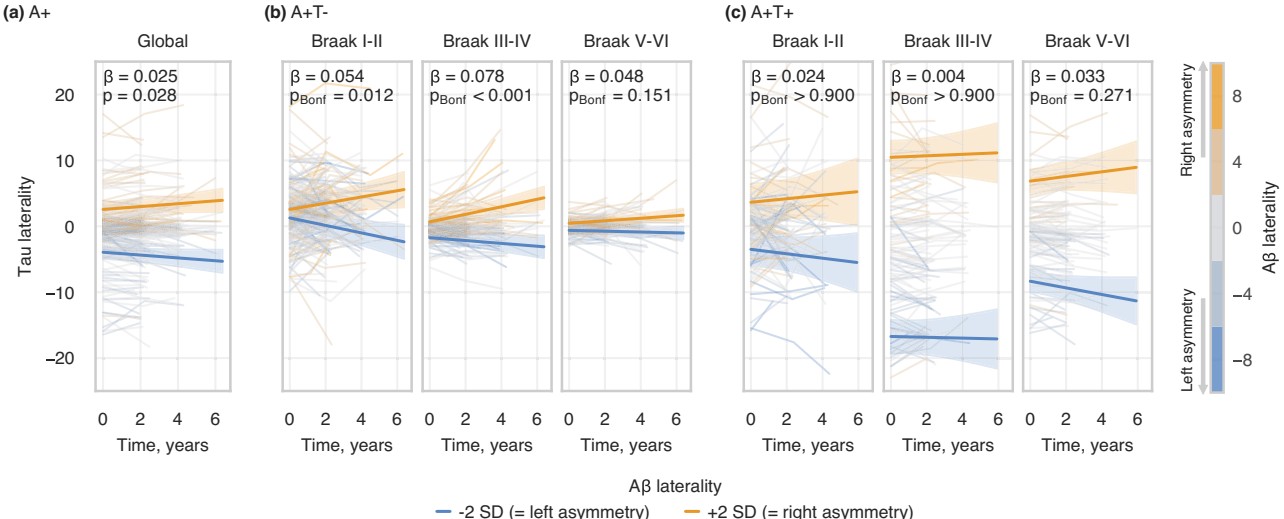

**Fig. 5 | Longitudinal analysis of the association between baseline Aβ laterality and changes over time in tau laterality. a** Whole A+ sample at global meta-ROI (i.e., whole-brain for each hemisphere); **b** A+T- subsample at Braak meta-ROIs; **c** A+T+ subsample at Braak meta-ROIs. The statistical analyses were performed using linear mixed effects models with random intercepts and slopes for time and participants (tau LI ~ time * (age$_{baseline}$ + sex + Aβ LI$_{baseline}$) + [1 + time | participant]), with p-values Bonferroni-corrected for the number of meta-ROIs tested in each subsample. The statistical annotations indicate the standardized effect size and significance level of the interaction between time and baseline Aβ laterality on tau laterality. For visualisation, regression lines represent the modelled mean tau laterality with 95% confidence intervals, plotted for LI$_{ref}$ ± 2 SD of baseline Aβ laterality, where LI$_{ref}$ = 0 (i.e., perfect Aβ symmetry). Colorbar indicates baseline Aβ laterality index. Aβ amyloid-beta, LI laterality index.

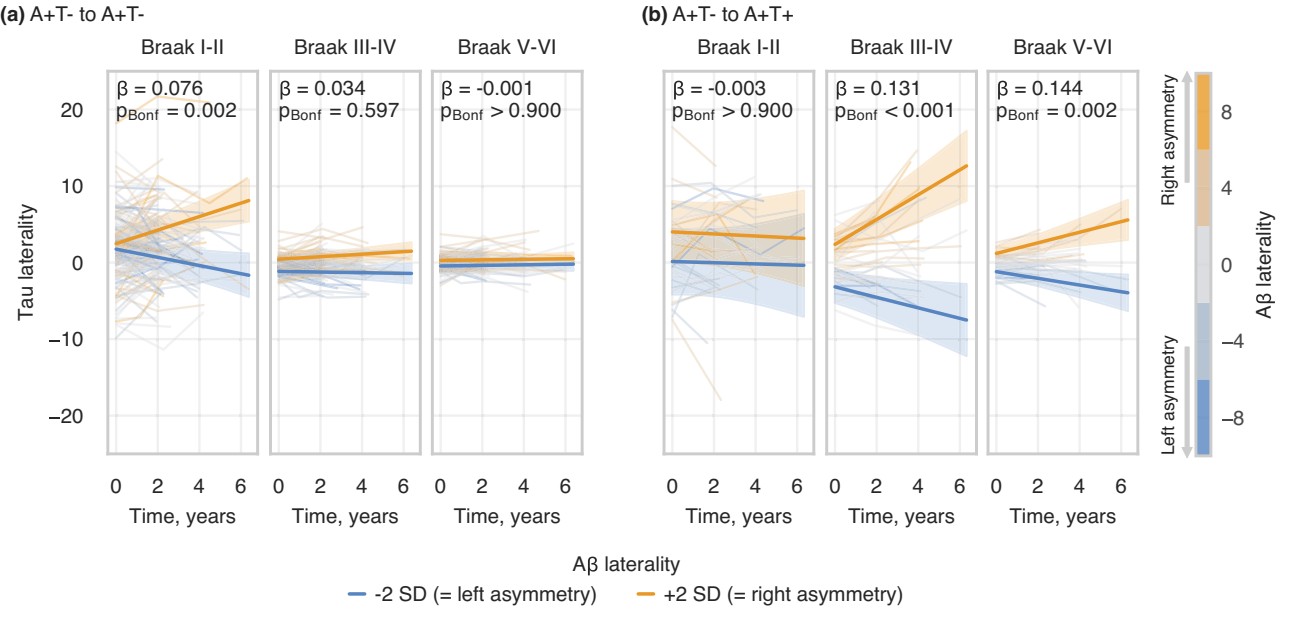

**Fig. 6 | Longitudinal analysis of the association between baseline Aβ laterality and changes over time in tau laterality at Braak meta-ROIs, stratified by conversion to T+ status. a** A+T- subsample who stay A+T- throughout their follow-up; **b** A+T- subsample who progress to A+T+ during their follow-up. The statistical analyses were performed using linear mixed effects models with random intercepts and slopes for time and participants (tau LI ~ time * (age$_{baseline}$ + sex + Aβ LI$_{baseline}$) + [1 + time | participant]), with p-values Bonferroni-corrected for the number of meta-ROIs tested in each subsample. The statistical annotations indicate the standardized effect size and significance level of the interaction between time and baseline Aβ laterality on tau laterality. For visualisation, regression lines represent the modelled mean tau laterality with 95% confidence intervals, plotted for LI$_{ref}$ ± 2 SD of baseline Aβ laterality, where LI$_{ref}$ = 0 (i.e., perfect Aβ symmetry). Colorbar indicates baseline Aβ laterality index. Aβ amyloid-beta, LI laterality index.

detectable with PET[26]. Instead, the local replication of pathogenic tau aggregates may play a more central role in controlling the overall rate of accumulation[26]. This process is likely influenced by many factors including, but not limited to, Aβ pathology, which increases local soluble tau levels, and hence can accelerate the tau aggregation process[24,25,33,34] or possibly other mechanisms, such as impairments in brain clearance, which could affect the accumulation rate of both pathologies[35,36].

Our results suggest that local Aβ pathology plays a critical role in determining asymmetry in tau accumulation, since tau asymmetry was preceded by asymmetry in Aβ. We systematically investigated whether this association could be explained by possible specificities in our data

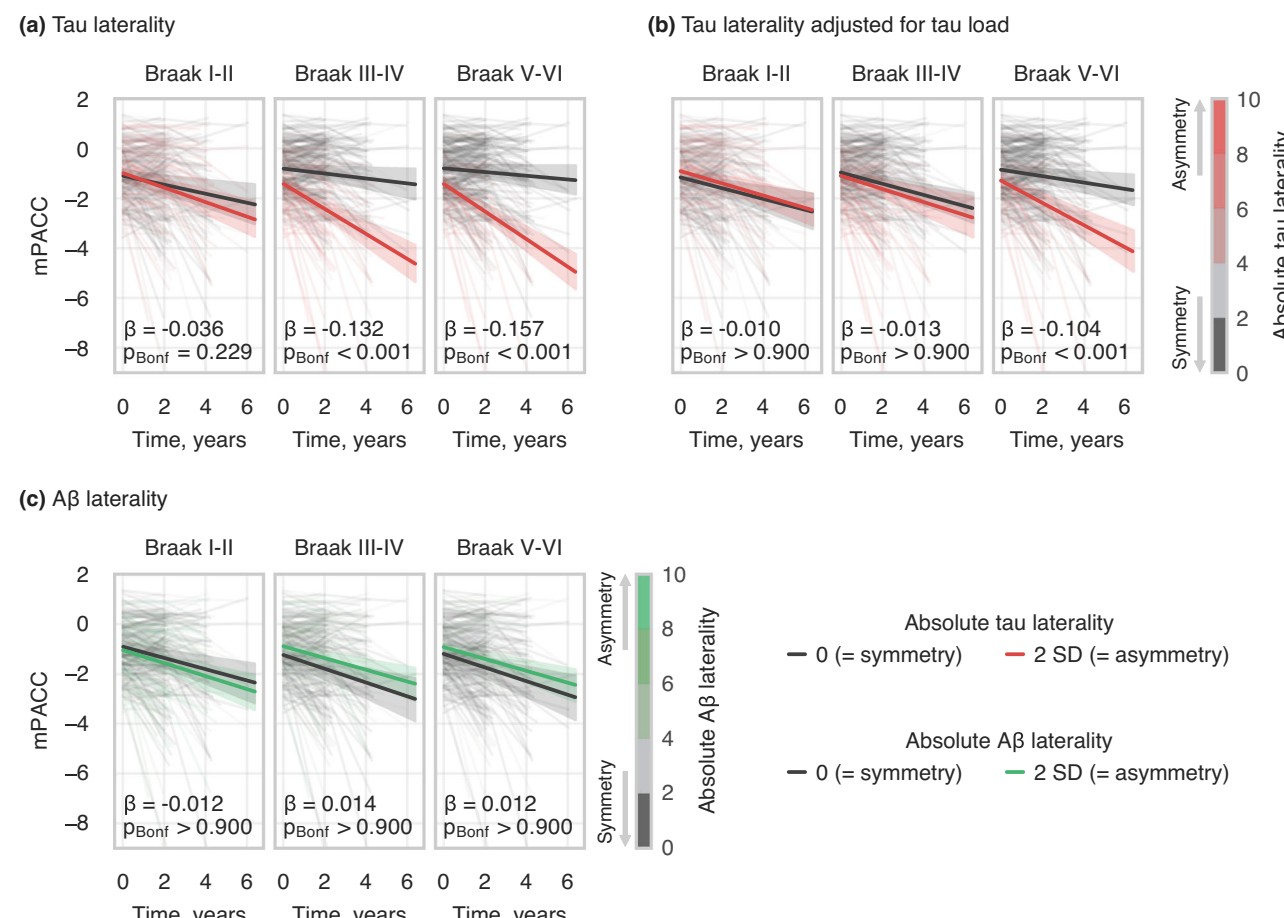

**Fig. 7 | Longitudinal analysis within A+ sample over Braak meta-ROIs predicting mPACC score over time with. a** Absolute baseline tau laterality; **b** Absolute baseline tau laterality after adjusting for tau load; **c** Absolute baseline Aβ laterality after adjusting for tau laterality, tau load, and Aβ load. The statistical analyses were performed using linear mixed effects models with random intercepts and slopes for time and participants (e.g., model depicted in panel **a**: mPACC ~ time * ($age_{baseline}$ + sex + tau $LI_{baseline}$) + [1 + time | participant]). The statistical annotations indicate the standardized effect size and significance level of the interaction between time and baseline Aβ/tau laterality on cognitive test score, with p-values Bonferroni-corrected for the number of meta-ROIs tested in each model. For visualisation, regression lines represent the modelled mean cognitive test score with 95% confidence intervals, plotted for $LI_{ref}$ and $LI_{ref}$ + 2 SD of baseline Aβ/tau laterality, where $LI_{ref}$ = 0 (i.e., perfect Aβ/tau symmetry). Aβ amyloid-beta, mPACC modified Preclinical Alzheimer Cognitive Composite, LI laterality index.

or analysis strategies[37]. However, laterality patterns in tau and Aβ were strongly associated also in three replication cohorts, which included participants at different stages along the AD continuum, had varying sample sizes, and used different PET tracers. Moreover, recent investigations have confirmed the lack of affinity of tau-PET tracers to Aβ plaques, suggesting that the type of tau-PET tracer would unlikely explain the associations found[38–40]. Another possible confounder could have been a systematic hemispheric difference in blood flow leading to an artificial difference in both Aβ and tau tracer uptake between hemispheres. However, we did not find evidence pointing in this direction. Furthermore, the association of lateralized patterns in Aβ and tau was independent from hemispheric differences in patterns of atrophy, indicating that laterality in brain atrophy did not bias the positive relationship between the two. Additional analyses using a different methodological approach (i.e., partial volume correction) further confirmed the robustness of the relationship between tau and Aβ asymmetries.

Beyond this cross-sectional relationship, the temporal dynamics of this association provide evidence for a possible mechanistic link. It is known that Aβ pathology starts to accumulate 10–30 years before cognitive impairment and precedes tau accumulation in AD[6,24]. Most importantly, our longitudinal analyses revealed that Aβ-positive subjects with a greater degree of Aβ asymmetry at their first timepoint developed more pronounced asymmetric tau distribution over time. The relationship between Aβ and tau laterality was particularly evident in A+T- subjects, who showed strong interaction between baseline Aβ laterality and progression of tau asymmetry over time, especially in areas corresponding to early and intermediate Braak stages (I-II and III-IV). In contrast, a strong association between baseline tau and Aβ asymmetry was found in A+T+ subjects, but no association between Aβ laterality and progression over time of tau toward a more asymmetric distribution was seen. Interestingly, A+T- individuals who did not show progression in neocortical tau pathology during their follow-up exhibited a link between asymmetrical Aβ deposition and increased tau laterality over time in the entorhinal cortex (i.e., Braak I-II), whereas those whose tau distribution extended to neocortical regions during the follow-up period displayed this interaction within neocortical areas associated with later disease stages (i.e., Braak III-IV and V-VI). These results further support a role of Aβ in determining the patterns of tau distribution from the early phases of the disease course. While asymmetric distribution of Aβ pathology has been previously reported in AD[41–45], recent evidence from patients with comorbid epilepsy demonstrated co-lateralisation of Aβ and tau in the epileptogenic hemisphere[46]. Our findings extend this observation, suggesting that this co-occurrence of Aβ and tau is not limited to specific comorbid

conditions, but represents a fundamental characteristic of AD pathophysiology.

As our findings suggest that asymmetric tau accumulation may be driven by hemispheric bias in Aβ deposits, we investigated to what extent the asymmetry in the distribution of pathology is related to cognitive decline. Importantly, asymmetric tau was observed in 32% (146/452) of A+T+ individuals, representing 17% (146/837) of the broader Aβ-positive cohort. While this subgroup does not constitute the majority of cases, its size (nearly one-third of those with elevated tau) underscores its clinical relevance. Consistent with previous studies investigating tau asymmetry[7,9,13,16–18], we found that left-predominant asymmetry (22%; 102/452) was more common than right-predominant asymmetry (10%; 44/452), and that asymmetric tau distribution was associated with worse cognitive decline over time. This association was largely driven by individuals exhibiting both higher average tau load and greater tau asymmetry. Nevertheless, greater asymmetry in tau deposition in late Braak stage regions was associated with steeper cognitive decline, even after adjusting for tau burden. In contrast, Aβ laterality did not have any effect on cognition. Together, these findings suggest that tau lateralisation in brain regions mainly affected in advanced disease stages might be associated with faster cognitive decline.

There are a few limitations to this study that should be considered. First, to provide even stronger causal roles of connectivity or Aβ deposition in determining the spatial distribution of tau pathology more data is needed from large-scale cohorts. Preferably, participants should be followed longitudinally over 10–20 years with MRI and PET scans starting many years prior to both Aβ and tau accumulation until development of substantial levels of each of the pathologies, but such cohorts are not yet available. Nevertheless, our analyses provided compelling evidence for a pivotal role of Aβ in determining the pattern of tau accumulation. Moreover, the results of the clinical trial for lecanemab, an Aβ-targeting treatment, has shown that removal of Aβ pathology leads to a reduced increase in tau PET signal over time[47], indicating a direct causal relationship between Aβ plaque pathology and tau accumulation. Second, as our null-finding regarding the association between inter-hemispheric connectivity and asymmetry in tau accumulation were based on macro-scale analyses, future work may benefit from assessing this relationship at finer spatial scales and employing more sophisticated approaches[48], which could be the key to understanding whether prion-like tau spread via connectivity drives lateralisation of tau pathology. Furthermore, tau accumulation and propagation likely lead to reduction in connectivity, which could have obscured the effect we were trying to investigate. Again, the full extent of the associations between connectivity and tau could be investigated only with an extensive longitudinal study following participants converting across the different stages of the AD continuum from A-T- to A+T+. A third limitation is the relatively small number of subjects with atypical AD and asymmetrical tau distribution. For example, we cannot rule out that a larger sample size could have allowed us to uncover more subtle differences in structural or functional connectivity. Finally, our cohort does not include symptom duration data, precluding analysis of whether group differences in tau burden reflect disease stage or are characteristic to tau lateralisation. However, we addressed this by adjusting for tau load in our statistical models, which serves as a proxy of clinical disease severity and helps distinguish stage-related effects from lateralisation-specific processes.

In summary, our study suggests that regional hemispheric vulnerability to AD pathology, especially Aβ deposits, might play a critical role in determining asymmetric distribution of tau. Specifically, asymmetric Aβ deposition appears to precede and is related to asymmetric tau accumulation, indicating that Aβ plays a critical role in the early pathophysiological cascade of AD by the suggested co-localisation of the two proteins. These results strengthen the links between Aβ and tau, supporting the hypothesis that early intervention

with anti-amyloid treatments[49–51] could help to limit the accumulation of tau pathology and downstream cognitive decline. However, further research is needed to identify the underlying mechanisms regarding the cause of one hemisphere being more susceptible to initial Aβ aggregation resulting in pathological asymmetry.

## Methods
### Main cohort
All participants within the main cohort were part of the Swedish BioFINDER-2 cohort (NCT03174938) and provided informed consent prior to participation. The study was approved by the Swedish Ethical Review Authority, and the data was collected according to the Declaration of Helsinki. The inclusion criteria for the present study were (1) evidence of Aβ pathology (A+); (2) age > 50 years; (3) did not fulfil the clinical criteria for other neurodegenerative diseases besides AD (e.g., frontotemporal dementia or Parkinson's disease); (4) no other known severe neurological condition (e.g., brain tumour); (5) have at least one tau-PET scan. This resulted in a population of 837 participants.

In detail, cognitive assessments included mini–mental state examination (MMSE) and modified preclinical Alzheimer cognitive composite (mPACC) scores. Both clinically unimpaired and impaired individuals were included if they met the previously mentioned criteria (see Supplementary S1 for detailed inclusion and exclusion criteria). Aβ positivity was defined using a previously established cut-off (SUVR > 1.033) determined via Gaussian mixture modelling (GMM) applied on the Aβ-PET uptake in a neocortical composite meta-ROI (regions detailed in Table S1.1), based on data from both the cognitively unimpaired and impaired individuals in the BioFINDER-2 cohort[52]. In case Aβ-PET was not available (i.e., all the AD patients in the dementia stage, by study design), Aβ status was determined using CSF Aβ42/40 ratio measurements, with either the Roche Elecsys assay (cutoff = 0.080) or, if that was not possible, the Lumipulse G immunoassay (cutoff = 0.072)[25,53].

### Hemispheric asymmetry of pathology
For each participant, hemispheric laterality index (LI) of Aβ and tau pathologies was calculated for all brain regions and meta-ROIs using the following equation: LI (%) = 100 × (right SUVR - left SUVR) / (right SUVR + left SUVR)[10,16,54]. The cross-sectional subsample of 475 participants included only A+ individuals who were also tau positive (T+) as defined by tau-PET uptake in a temporal meta-ROI (i.e., Braak stages I-IV regions; detailed in Table S1.1). The T+ threshold (SUVR > 1.362) was derived in the BioFINDER-2 cohort combining two approaches for cut-off estimation: 2 SD above the mean of Aβ-negative cognitively unimpaired older adults and GMM of the full cohort[27,28].

These participants were then categorised into three groups based on the spatial distribution of tau pathology using the temporal meta-ROI tau LI. Participants with LI exceeding ±1 SD from perfect symmetry (i.e., $LI_{ref} = 0$) were assigned into asymmetric groups – left tau asymmetric (LA) if LI < $LI_{ref}$ − 1 SD, tau symmetric (S) if |LI| < $LI_{ref}$ + 1 SD, or right tau asymmetric (RA) if LI > $LI_{ref}$ + 1 SD; subjects that had LI within the ±5% range of the threshold were dropped. The resulting thresholds were: |LI| > 9.70 for asymmetry and |LI| < 8.78 for symmetry. Of the 452 A+T+ subjects, 352 underwent diffusion MRI (dMRI), 318 resting-state functional MRI (RSfMRI), and 233 Aβ-PET scans. Furthermore, a second longitudinal subsample of 289 A+ participants with at least two available tau-PET scans and Aβ-PET scan at their first timepoint (i.e., baseline) were included. This subsample was stratified into A+T- (n = 180) and A+T+ (n = 109) based on the tau positivity at baseline (see Supplementary Table S2.2 for demographics).

All statistical analyses were performed using tableone and statsmodels packages in Python[55,56]. Brain surface plots were created using BrainSpace toolbox in Python[57]. Demographics were compared between the tau asymmetry groups using either one-way ANOVA,

Kurskal-Wallis, or Chi-squared test depending on the type and distribution of the data. Comparisons of tau load between the groups were performed using ordinary least squares (OLS) multiple linear regressions (OLS: tau load ~ age + sex + group). The significance level (p < 0.05) was Bonferroni-corrected for the number of group comparisons performed (3 comparisons: LA vs S, RA vs S, and LA vs RA).

## Image acquisition

**PET.** All study participants underwent PET scans on a digital GE Discovery MI scanner (General Electric Medical Systems). On this platform, tau-PET using [18F]RO948 (70–90 min after the injection of $365 \pm 20$ MBq) and Aβ-PET using [18F]flutemetamol (90–110 min after the injection of ~185 MBq) were conducted[58]. For tau-PET, SUVR maps were calculated using the inferior cerebellar cortex as reference region[59]. For Aβ-PET, a cortical composite SUVR was calculated using whole cerebellum as the reference region[60]. Mean SUVR values were extracted for each region of the Desikan-Killiany atlas after registering the PET images to the corresponding MRI T1-weighted scan. The average SUVR values were also calculated for meta-ROIs: global (i.e., whole brain), temporal (i.e., Braak I-IV), Braak I-II, Braak III-IV, Braak V-VI, Early-Aβ, Intermediate-Aβ, and Late-Aβ (Fig. 8; see Supplementary Table S1.1 for detailed overview)[4,5,61].

**MRI protocol.** The MRI imaging was conducted using a MAGNETOM Prisma 3 T MRI scanner (Siemens Healthineers, Forchheim, Germany) with a 64-channel head/neck coil. RSfMRI was acquired using a gradient-echo planar sequence (eyes closed; in-plane

resolution = $3 \times 3$ mm$^2$; slice thickness = 3.6 mm; repetition time = 1020 ms; echo time = 30 ms; flip-angle = 63°; 462 dynamic scans over a period of 7.85 min). For dMRI, 104 diffusion-weighted imaging volumes were acquired using a single-shot echo-planar imaging sequence (repetition time = 3500 ms; echo time = 73 ms; resolution = $2 \times 2 \times 2$ mm$^3$; field of view = $220 \times 220 \times 124$ mm$^3$; b-values range = 0, 100, 1000 and 2500 s/mm$^2$ distributed over 2, 6, 32 and 64 directions; 2-fold parallel acceleration and partial Fourier factor = 7/8). T1-weighted structural images were also acquired using a magnetisation-prepared rapid gradient-echo (MPRAGE) sequence (inversion time = 1100 ms; flip-angle = 9°; echo time = 2.54 ms; echo spacing = 7.3 ms; repetition time = 1900 ms; receiver bandwidth = 220 Hz/pixel; voxel size = $1 \times 1 \times 1$ mm$^3$). Generalised autocalibrating partially parallel acquisitions (GRAPPA) was applied with an acceleration factor of 2 and 24 reference lines. Additionally, ASL scans were acquired on a subset of the sample using a prototype 3D pseudo-continuous (pCASL) sequence with background suppression and gradient- and spin-echo (GRASE) readout (TR/TE = 4600/21.76 ms, labelling duration = 1500 ms, post-labelling delay = 2000 ms, voxel size = $1.9 \times 1.9 \times 4$ mm$^3$, 12 control/label pairs, scan time = 5:50 min) with proton density (M0) image (TR = 4000 ms) also acquired for calibration[62].

**MRI pre-processing.** T1-weighted structural MPRAGE images were pre-processed using FreeSurfer (version 6.0, https://surfer.nmr.mgh.harvard.edu)[63]. This included various steps, such as correction for intensity homogeneity, skull stripping, and tissue segmentation.

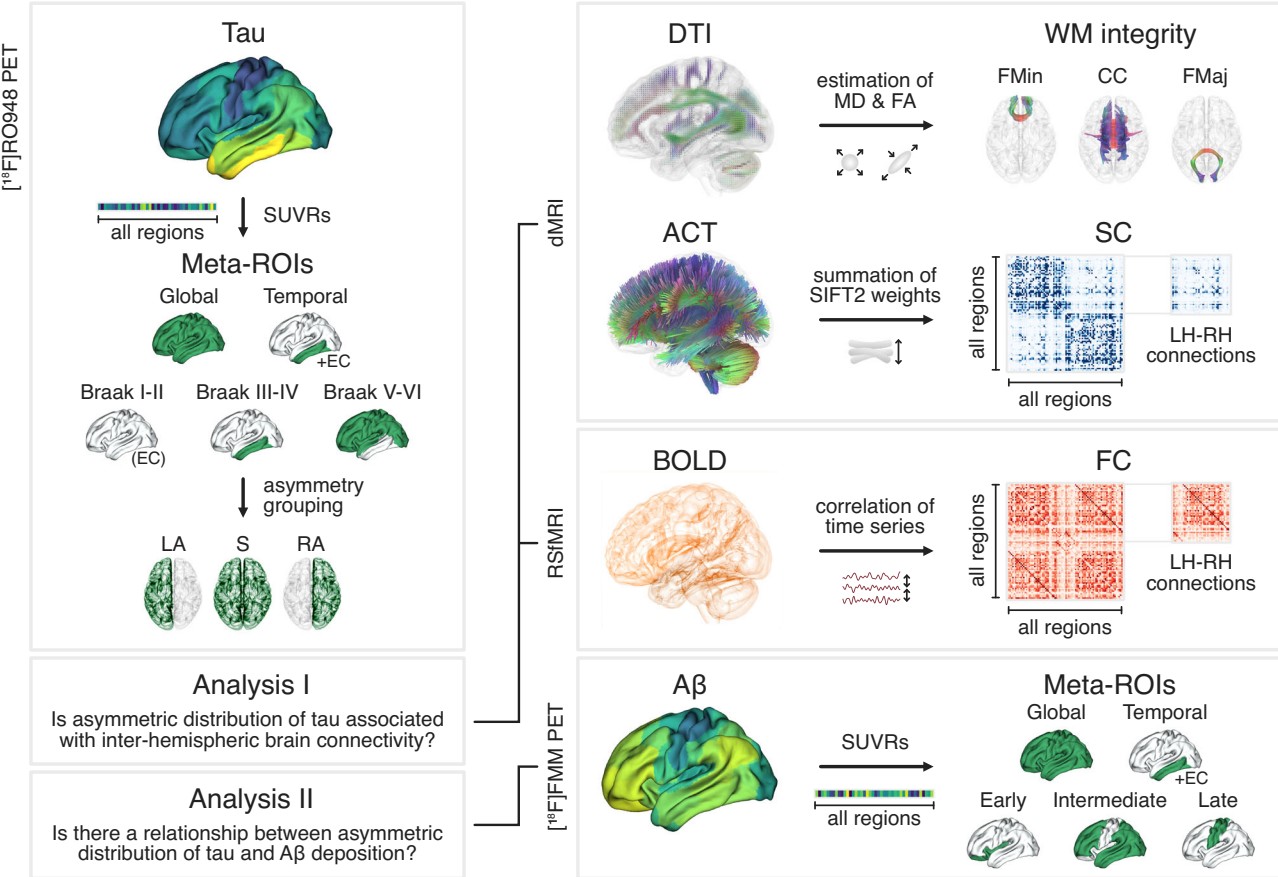

**Fig. 8 | An overview of the data processing steps and analyses.** PET positron emission tomography, dMRI diffusion magnetic resonance imaging, RSfMRI resting-state functional magnetic resonance imaging, FMM flutemetamol, SUVR standardized uptake value ratio, Meta-ROI meta region of interest, EC entorhinal cortex, LA left tau asymmetric, S tau symmetric, RA right tau asymmetric, DTI diffusion tensor imaging, MD mean diffusivity, FA fractional anisotropy, FMin Forceps Minor, CC Corpus Callosum, FMaj Forceps Major, ACT anatomically constrained tractography, SIFT2 spherical-deconvolution informed filtering of tractograms, LH/RH left/right hemisphere, SC structural connectivity, BOLD blood-oxygen-level-dependent imaging, FC functional connectivity, Aβ amyloid-beta.

Diffusion MRI images were pre-processed using a combination of FSL (FMRIB Software Library, version 6.0.4; Oxford, UK) and MRtrix3 tools[64,65]. These images underwent correction for susceptibility-induced distortions, using images acquired with opposite phase polarities, motion, and Eddy current-induced artifacts. RSfMRI pre-processing utilised the Configurable Pipeline for the Analysis of Connectomes (C-PAC), with additional tools including AFNI and ANTs[66–68]. Processing included slice timing correction, T2-based unwarping for susceptibility distortion correction, and spatial normalisation to MNI152 standard space using nonlinear registration. Nuisance regression removed white matter (WM) and cerebrospinal fluid (CSF) signals via mean regression, along with 24 motion parameters (6 rigid-body + temporal derivatives) and linear trends. Data were bandpass filtered (0.01–0.1 Hz), and outlier frames were censored using a DVARS threshold (>75th percentile + 1.5 × IQR). Participants exceeding mean/max frame-wise displacement thresholds (0.7/3.0 mm) were excluded. No spatial smoothing was applied.

**Estimation of MRI-based connectivity measures.** Structural connectivity (SC) and functional connectivity (FC) were estimated using a combination of MRtrix3, FSL, FreeSurfer, and Nilearn software packages, integrated through NiPype interfaces in Python[63–65,69,70]. See Supplementary S1 for a more detailed description of the process.

In brief, FC matrices were derived from pre-processed subject-space RSfMRI data using Pearson correlation after Fisher's z-transformation[71], and estimated for 84 cortical and sub-cortical regions defined in the Desikan-Killiany atlas and in the ASEG protocol[72,73]. For SC, response functions for grey matter (GM), WM and CSF were estimated using the "dhollander" algorithm[74] on pre-processed diffusion MRI data from 60 cognitively unimpaired A-T- and 40 A+T- participants of the BioFINDER-2 cohort, and fibre orientation distributions (FOD) were derived via multi-shell multi-tissue constrained spherical deconvolution[75]. Anatomically-constrained tractography (ACT) with the "iFOD2" algorithm generated ten million streamlines, which were filtered using SIFT2 to reduce overestimation bias[76–79]. SC matrices were constructed for the same 84 regions employed for FC[72], with summation of SIFT2-weighted streamlines per region used as edges.

For evaluating microstructural integrity, the diffusion tensor imaging (DTI) model was fitted to the diffusion MRI data using the volumes acquired with a b-value up to 1000 s/mm², and fractional anisotropy (FA) and mean diffusivity (MD) maps were derived[80,81]. The main inter-hemispheric white matter tracts – corpus callosum, forceps major, forceps minor – were segmented using TractSeg and the mean FA and MD values were extracted for each tract[82].

### Association between brain connectivity and asymmetric distribution of tau pathology

**Inter-hemispheric connectivity and microstructural integrity.** To reduce noise and focus on the most biologically plausible connections, we masked both the FC and SC matrices for all subjects by selecting only the top 10% inter-hemispheric connections identified in 294 age-matched healthy controls with no evidence of Aβ or tau pathology from the BioFINDER-2 cohort (see Supplementary S1 for more information).

For each subject, we then calculated the global inter-hemispheric FC and SC by averaging the connections between all the left nodes connecting to the right (Fig. 8). The associations between global absolute tau laterality and these connectivity averages were then investigated in the cross-sectional A+T+ sample using OLS multiple linear regressions (OLS: FC/SC ~ age + sex + average global tau load + absolute global tau LI). Subsequently, these connectivity measures were compared between the tau asymmetry groups (i.e., LA vs S, RA vs S, and LA vs RA; three OLS models: FC/SC ~ age + sex + average global

tau load + group) with the significance level (p < 0.05) Bonferroni-corrected for 3 comparisons. Parallel analyses assessed FA and MD in inter-hemispheric tracts using identical models.

Edge-wise associations between homotopic inter-hemispheric FC and SC (n = 36 connections) and tau pathology were analysed using linear regressions. For each region, two models were tested to assess the relationships with (1) absolute tau laterality (OLS: homotopic FC/SC ~ age + sex + bilateral tau load + absolute tau LI) and (2) bilateral tau burden (OLS: homotopic FC/SC ~ age + sex + bilateral tau load). False Discovery Rate (FDR) correction (Benjamini-Hochberg method, p < 0.05) was applied across all 36 connections for both analyses.

**Whole-brain connectomics.** Network Based Statistic (NBS) method[83] was applied on FC and SC matrices to compare the whole-brain connectome differences between the tau asymmetry groups. The primary NBS analyses were performed on the masked matrices (retaining the top 10% strongest connections from healthy controls; see Supplementary S1 for more information on masking) to focus on biologically plausible connections. In short, NBS is a cluster-based method that consists of three main steps: (1) edges that, when compared between groups, surpass a given statistical threshold (e.g., t = 3.0) are identified, (2) components (i.e., connected subgraphs or clusters of topologically contiguous supra-threshold edges) are detected, and (3) permutation testing adjusting for family-wise error (FWE) is performed to assign a p-value for each detected component based on its size relative to the null distribution of component sizes obtained through permutation. It is often recommended to repeat NBS using different statistical thresholds in step (1)[83]. This study used thresholds of 2.5, 3.0, and 3.5. Moreover, the comparisons were adjusted for age, sex, and average global tau load. 5000 permutations were performed, and the significance level was set to p < 0.05 after FWE correction. Sensitivity analyses were additionally performed on the unthresholded connectivity matrices to ensure that the masking approach did not bias the results.

### Association between the distribution of Aβ and tau pathologies

The association between the laterality of Aβ and tau (i.e., Aβ LI vs tau LI) was investigated in the cross-sectional A+T+ sample for each region defined in the Desikan-Killiany atlas and each meta-ROI described above (Fig. 8). The statistical analyses were performed using OLS multiple linear regressions (OLS: tau LI ~ age + sex + Aβ LI), with the significance level (p < 0.05) Bonferroni-corrected for the number of meta-ROIs the analysis was performed on or FDR-corrected for region-specific analyses across all regions.

The association between baseline Aβ laterality and tau laterality over time was investigated in the longitudinal A+, A+T-, and A+T+ samples for global, Braak I-II, Braak III-IV, and Braak V-VI meta-ROIs. The statistical analyses were performed using Linear Mixed Effects (LME) models with random intercepts and slopes for time and participants (LME: tau LI ~ time * (age$_{baseline}$ + sex + Aβ LI$_{baseline}$) + [1 + time | participant]), with the significance level (p < 0.05) Bonferroni-corrected for the number of meta-ROIs the analysis was performed on.

### Association between the distribution of pathologies and cognition

The association between baseline Aβ and tau laterality and cognitive decline was investigated in the longitudinal A+ sample for Braak I-II, Braak III-IV, and Braak V-VI meta-ROIs. The statistical analyses were performed similarly using LME modelling with random intercepts and slopes for time and participants, but with three different models: (1) a base model testing tau laterality effects on cognition (LME: mPACC ~ time * (age$_{baseline}$ + sex + tau LI$_{baseline}$) + [1 + time | participant]), (2) a model testing tau laterality effects on cognition but controlling for tau load (LME: mPACC ~ time * (age$_{baseline}$ + sex + tau load$_{baseline}$ + tau

LI$_{baseline}$) + [1 + time | participant]), and (3) a model testing Aβ laterality effects on cognition (LME: mPACC ~ time * (age$_{baseline}$ + sex + Aβ LI$_{baseline}$) + [1 + time | participant]). The significance level (p < 0.05) for all models was Bonferroni-corrected for the number of meta-ROIs the analysis was performed on.

## Replication in independent cohorts

Three external cohorts (see Table S2.9 for demographics) were used to validate the association between the laterality of Aβ and tau distribution – Open Access Series of Imaging Studies (OASIS-3; https://sites.wustl.edu/oasisbrains/), Anti-Amyloid Treatment in Asymptomatic Alzheimer's Disease (A4; https://www.a4studydata.org/), and Alzheimer's Disease Neuroimaging Initiative (ADNI; https://adni.loni.usc.edu/). For each replication cohort, only subjects with evidence of both Aβ and tau (A+T+) and with at least one Aβ-PET and one tau-PET scan available were selected.

Aβ positivity was pre-defined in a cohort-specific manner for OASIS-3 and ADNI datasets. OASIS-3 used [11 C]PiB (SUVR > 1.42) and [$^{18}$F]florbetapir (SUVR > 1.19) tracers, as described previously[84]. ADNI used [$^{18}$F]florbetaben (SUVR > 1.08) and [$^{18}$F]florbetapir (SUVR > 1.11) tracers[85]. A4 used [$^{18}$F]florbetapir PET for Aβ imaging and the ADNI-defined cut-off (SUVR > 1.11) was applied to determine Aβ status. Tau imaging across all replication cohorts used [$^{18}$F]flortaucipir PET. Tau positivity was defined as Temporal meta-ROI SUVR > 1.34, a threshold established in prior work based on 2 SD from the mean tau-PET uptake within cognitively unimpaired Aβ-negative elderly in multiple combined cohorts[86].

## Reporting summary

Further information on research design is available in the Nature Portfolio Reporting Summary linked to this article.

# Data availability

Four different cohorts were used in this study: BioFINDER-2, ADNI, A4, and OASIS-3. For BioFINDER-2 data, anonymized data will be shared by request from a qualified academic investigator as long as data transfer is in agreement with European Union legislation on the General Data Protection Regulation and decisions by the Swedish Ethical Review Authority and Region Skåne, which should be regulated in a material transfer agreement. ADNI, A4 and OASIS-3 are publicly available datasets and can be obtained from http://adni.loni.usc.edu/, https://ida.loni.usc.edu/ and https://sites.wustl.edu/oasisbrains/, respectively. Source data are provided with this paper.

# Code availability

The analysis code is available at Zenodo and is maintained on GitHub[87].

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

## Acknowledgements

We would like to express our gratitude to the research volunteers who participated in the studies from which these data were obtained and their supportive families. The work of the Swedish BioFINDER group is supported by European Research Council (ADG-101096455), Alzheimer's Association (ZEN24-1069572, SG-23-1061717), GHR Foundation, Swedish Research Council (2018-02052, 2021-02219, 2022-00775), ERA PerMed (ERAPERMED2021-184), Knut and Alice Wallenberg foundation (2022-0231), Strategic Research Area MultiPark (Multidisciplinary Research in Parkinson's disease) at Lund University, Swedish Alzheimer Foundation (AF-980907, AF-994229, AF-1011799), Swedish Brain Foundation (FO2021-0293, FO2023-0163), Lilly Research Award Program, Michael J Fox Foundation (MJFF-025507), Parkinson foundation of Sweden (1412/22), Cure Alzheimer's fund, Rönström Family Foundation, Kockska Foundations, Berg Family Foundation, Greta and Johan Kock Foundation, Konung Gustaf V:s och Drottning Victorias Frimurarestiftelse, Skåne University Hospital Foundation (2020-O000028), Regionalt Forskningsstöd (2022-1259),WASP and DDLS Joint call for research projects (WASP/DDLS22-066), and Swedish federal government under the ALF agreement (2022-Projekt0080, 2022-Projekt0107). The authors wish to acknowledge Dr. Josef Pfeuffer at Siemens Healthineers AG, Erlangen, Germany, for providing the pcASL research sequence. The precursor of [18F]flutemetamol was sponsored by GE Healthcare. The precursor of [18F]RO948 was provided by Roche. The funding sources had no role in the design and conduct of the study; in the collection, analysis, interpretation of the data; or in the preparation, review, or approval of the manuscript. The A4 study is funded by a public-private-philanthropic partnership, including funding from the NIH-National Institute on Aging, Eli Lilly and Co., Alzheimer's Association, Accelerating Medicines Partnership, GHR Foundation, an anonymous foundation, and additional private donors, with in-kind support from Avid and Cogstate. The ADNI was launched in 2003 as a public-private partnership, led by Principal Investigator Michael W. Weiner, MD. A full list of consortium members appears in Supplementary Note 1. Data collection and sharing for the ADNI is funded by the National Institute on Aging (National Institutes of Health Grant U19AG024904). The grantee organisation is the Northern California Institute for Research and Education. In the past, ADNI has also received funding from the National Institute of Biomedical Imaging and Bioengineering, the Canadian Institutes of Health Research, and private sector contributions through the Foundation for the National Institutes of Health (FNIH) including generous contributions from the following: AbbVie, Alzheimer's Association; Alzheimer's Drug Discovery Foundation; Araclon Biotech; BioClinica, Inc.; Biogen; Bristol-Myers Squibb Company; CereSpir, Inc.; Cogstate; Eisai Inc.; Elan Pharmaceuticals, Inc.; Eli Lilly and Company; EuroImmun; F. Hoffmann-La Roche Ltd and its affiliated company Genentech, Inc.; Fujirebio; GE Healthcare; IXICO Ltd.; Janssen Alzheimer Immunotherapy Research & Development, LLC.; Johnson & Johnson Pharmaceutical Research & Development LLC.; Lumosity; Lundbeck; Merck & Co., Inc.; Meso Scale Diagnostics, LLC.; NeuroRx Research; Neurotrack Technologies; Novartis Pharmaceuticals Corporation; Pfizer Inc.; Piramal Imaging; Servier; Takeda Pharmaceutical Company; and Transition Therapeutics.

## Author contributions

T.E.A., N.S and O.H. designed the study. T.E.A. and N.S. had full access to raw data. T.E.A. performed data processing and carried out the statistical analyses. T.E.A. wrote the manuscript and had the final responsibility to submit it for publication. N.S. and O.H. contributed as the main supervisors of the work. All other authors (R.O., R.S., A.P.B., L.E.J., H.H.B., J.R., L.K., K.A., O.S., D.W., J.W.V., E.S., S.P., N.M.C.) contributed demographic, clinical, biomarker, and neuroimaging data, contributed to the interpretation of the results, and critically reviewed the manuscript.

## Funding

## Competing interests

O.H. is an employee of Eli Lilly and Lund University. R.S. has received consultancy/speaker fees from Eli Lilly, Novo Nordisk, Roche and Triolab. S.P. has acquired research support (for the institution) from ki elements / ADDF and Avid. In the past 2 years, he has received consultancy/speaker fees from Bioartic, Esai, Eli Lilly, Novo Nordisk, and Roche. N.M.C. has received consultancy/speaker fees from Biogen, Eli Lilly, Owkin, and Merck. The precursor of [18F]flutemetamol was sponsored by GE Healthcare. The other authors declare no competing interests.

## Additional information

[1]Clinical Memory Research Unit, Department of Clinical Sciences Malmö, Faculty of Medicine, Lund University, Lund, Sweden. [2]Alzheimer Center Amsterdam, Neurology, Vrije Universiteit Amsterdam, Amsterdam UMC, Amsterdam, the Netherlands. [3]Neurodegeneration, Amsterdam Neuroscience, Amsterdam, the Netherlands. [4]Memory Clinic, Skåne University Hospital, Malmö, Sweden. [5]Department of Physiology and Pharmacology, Université de Montréal, Montréal, Quebec, Canada. [6]Centre de Recherche de l'Institut Universitaire de Gériatrie de Montréal, Montréal, Quebec, Canada. [7]Radiology and Nuclear Medicine, Vrije Universiteit Amsterdam, Amsterdam UMC, Amsterdam, the Netherlands. [8]Brain Imaging, Amsterdam Neuroscience, Amsterdam, the Netherlands. [9]SciLifeLab, Department of Clinical Sciences Malmö, Faculty of Medicine, Lund University, Lund, Sweden. [10]Department of Neuropsychology, Ruhr University Bochum, Bochum, Germany. [11]Diagnostic Radiology, Institution for Clinical Sciences, Lund University, Lund, Sweden. [12]Wallenberg Center for Molecular Medicine, Lund University, Lund, Sweden. [25]These authors jointly supervised this work: Nicola Spotorno, Oskar Hansson.
✉e-mail: toomas_erik.anijarv@med.lu.se; nicola.spotorno@med.lu.se; oskar.hansson@med.lu.se

## the Alzheimer's Disease Neuroimaging Initiative

**Michael Weiner[13], Paul Aisen[14], Ronald Petersen[15], Clifford R. Jack Jr.[15], William Jagust[16], Susan Landau[16], Monica Rivera-Mindt[17], Ozioma Okonkwo[18], Leslie M. Shaw[19], Edward B. Lee[19], Arthur W. Toga[20], Laurel Beckett[21], Danielle Harvey[21], Robert C. Green[22], Andrew J. Saykin[23], Kwangsik Nho[23], Richard J. Perrin[24] & Duygu Tosun[13]**

[13]UC San Francisco, San Francisco, CA, USA. [14]University of Southern California, Los Angeles, CA, USA. [15]Mayo Clinic, Rochester, NY, USA. [16]UC Berkeley, Berkeley, CA, USA. [17]Mount Sinai School of Medicine, New York, NY, USA. [18]University of Wisconsin, Madison, WI, USA. [19]University of Pennsylvania, Philadelphia, PA, USA. [20]UC Los Angeles, Los Angeles, CA, USA. [21]UC Davis, Davis, CA, USA. [22]Boston University, Boston, MA, USA. [23]Indiana University, Bloomington, IND, USA. [24]Washington University St. Louis, St. Louis, MO, USA.

