## [Transparent Peer Review file · Nature Communications]

Hemispheric Asymmetry of Tau Pathology is Related to Asymmetric Amyloid Deposition in Alzheimer's Disease

Corresponding Author: Mr Toomas Erik Anijärv

Version 0:

Reviewer comments:

Reviewer #1

(Remarks to the Author)

This manuscript examined the asymmetry of brain tau deposition along the AD spectrum. The authors used tau PET in the BIOFINDER-2 COHORT supplemented by amyloid PET in some and CSF measures of amyloid in others to assure amyloid positivity. In brief, using a straightforward definition of asymmetry, they found that a substantial proportion of individuals showed asymmetry and they show convincing data that this asymmetry is related to asymmetry of beta-amyloid brain deposition and not to patterns of cortical connectivity. Further explorations showed that baseline brain amyloid asymmetry predicted evolving tau asymmetry, and that the findings replicated in additional cohorts.

This is an excellent study that addresses a question of major importance in the field of dementia research. The question of how amyloid and tau are related is fundamental, and the argument about how patterns of tau deposition may or may not reflect patterns of amyloid have been debated for a while. This manuscript provides compelling data linking the localization of brain amyloid to the localization of tau and in my opinion will be widely read and cited.

I do have a number of comments and suggestions that I believe would improve the paper as well as highlighting some of the limitations. Here they are in order of importance:

1. The results are strong in showing the amyloid-tau relationship. What's not as clear is how commonly this is seen. If I understand correctly, the BIOFINDER cohort began with 837 individuals who were amyloid+ which was then trimmed to 475 deemed to have elevated tau and 452 in whom asymmetry could be definitively addressed. Of these 146 were either left or right asymmetric; that is 17% of everyone with AD (defining AD as amyloid positive) or 31% of everyone with AD and elevated tau. This is an important group, but it's not the majority of people with AD. It is incumbent on the authors to point this out. It doesn't diminish the importance of the finding because it is a common one.
2. Similarly the authors are not clear about how the samples were selected for the replication studies (OASIS, A4, ADNI). I may have missed it (could not find this information in the supplement), but please explain if there was any sort of similar restrictions on who was in the analysis for each cohort and the total study sample as a proportion of available participants with amyloid and tau PET data.
3. Same issue for the longitudinal data depicted in figures 5 and 6. How were participants selected for inclusion and what proportion of participants with amyloid and tau data were included?
4. The findings in figure 3 are very interesting. The authors did a nice job of showing that the strength of the amyloid/tau correlation is not the same across all regions. However an important remaining question is how specific are the regional associations between amyloid and tau? That is, in a given ROI (for example inferior temporal) how different is the correlation between the asymmetry of tau in the inferior temporal lobe and the asymmetry of amyloid in the inferior temporal lobe as opposed to (for example) tau in the inferior temporal lobe and amyloid asymmetry in the precuneus (or whole hemisphere for that matter)? It would not be difficult to run this analysis and depict it for all ROIs using a correlation matrix. I think many people would be interested in this result and it has consequences for mechanistic understanding of what's happening.
5. I also wonder if the authors considered looking at a subgroup of nodes and edges in their connectivity analyses. Were the results the same if the authors confined their analyses to brain regions with the highest tau deposition or brain regions showing the greatest asymmetry?

Reviewer #2

(Remarks to the Author)

In this paper, Anijärvi et al. investigate the factors influencing the asymmetric distribution of tau pathology in Alzheimer's disease (AD). The authors explored two primary hypotheses: (1) whether this asymmetry is related to inter-hemispheric connectivity, and (2) whether it is associated with asymmetry in amyloid-beta (A β) distribution. Their findings indicate no significant association between tau asymmetry and inter-hemispheric functional or structural connectivity. However, they report a strong association with A β laterality, suggesting that the asymmetry of A β pathology may be a key driver of asymmetric tau accumulation. Furthermore, baseline A β laterality predicted longitudinal tau laterality. Overall, this well-written and informative paper provides valuable new insights into the mechanisms underlying the observed heterogeneity in AD. To further strengthen the study, the following aspects require additional clarification.

Methods:

1. Composite Region Definition and A β Threshold: For lines 378 and 412, please provide detailed information on the specific brain regions included in the composite measure. Additionally, clarify how the A β -PET neocortical uptake threshold of >1.033 was defined in the previous study. Specifically, was a two-Gaussian mixture model used? If so, what was the sample size used? Only controls were used to establish this threshold?
2. Temporal Meta-ROI Definition: Regarding lines 104 and 386, please provide more detailed information about the temporal meta-ROI. Specify the exact brain regions used to construct it and clearly define the cut-off criteria employed. Was this cut-off determined using the same tracer and data from the same study as the current analysis?
3. Lateralization Index (LI) Interpretation: On line 390, the statement that an absolute LI value <1 is considered symmetric is unclear. A symmetrical distribution should ideally have an LI of 0. Please clarify.
4. Final Lateralization Threshold: Following the 5% range mentioned on line 391, what is the final threshold used to categorize asymmetry?
5. Resting-State fMRI Preprocessing: The description of resting-state fMRI preprocessing on line 442 requires more detail. Please specify:
 - o Whether slice-timing correction was applied.
 - o The method used for co-registration to standard space.
 - o The type of noise regression performed.
 - o Whether cerebrospinal fluid (CSF) and white matter signals were removed. If so, was mean signal regression or Principal Component Analysis (PCA) used, and if PCA, how many components were regressed out?
 - o Whether linear trends and motion components were accounted for.
 - o Whether spatial smoothing was applied, and if so, the kernel size used.
6. Inter-Hemispheric Connectivity Analysis: For lines 139 and 470, please elaborate on the implications of keeping only 10% of inter-hemispheric connections. If the weighted degree of these connections is computed, do the resulting connections cover all brain regions in the parcellation? Are there specific regions with a disproportionately higher weighted degree for inter-hemispheric connections compared to others? Including a supplementary figure illustrating the weighted degree of inter-hemispheric connections would be highly beneficial.
7. Network-Based Statistic (NBS): On line 486, please clarify whether the NBS analysis was performed on a thresholded connectivity matrix. What the outcome is if the NBS analysis were applied to the unthresholded matrix.
8. Threshold Consistency Across Studies: Regarding line 524, please confirm whether the different studies used to compute the A β positivity threshold employed the same approach (i.e., a two-Gaussian mixture model) as the previous data referenced in this study.

Results:

9. Demographics Table: Is there a higher proportion of left-handed individuals in the right-asymmetric (RA) group compared to the left-asymmetric (LA) group?
10. Tau and Inter-Hemispheric Connectivity: When comparing individuals with and without tau pathology, are there observable differences in inter-hemispheric connectivity? While the study investigates the association between connectivity changes and tau laterality, it is important to first establish whether tau accumulation itself alters these connections or only affects intra-hemispheric connections. If a control group without tau is not available, NBS correlations with continuous SUVR load shows any association between tau and inter-hemispheric connections?
11. Contralateral Connections: On line 148, the reported number of 1412 connections between contralateral regions seems inconsistent with a parcellation of 42 regions, where one would expect a maximum of 42 homologous inter-hemispheric connections. Please clarify how these 1412 connections were defined. If the analysis did not specifically focus on these 42 homologous connections, exploring them might yield particularly interesting insights.
12. A β -Tau Association in Temporal Regions: On line 178, the strongest effect sizes between tau and A β are reported in temporal regions. Do the individuals in this cohort exhibit tau pathology beyond these temporal regions? It would be helpful to include the number of individuals in each Braak stage in the demographics table. The strong correlation in temporal

regions might be driven by the prevalence of individuals in earlier Braak stages. If individuals in Braak stages V-VI are specifically examined, is the effect size of the A β -tau association stronger within this temporal meta-ROI?

13. SILA Algorithm Explanation: The SILA algorithm mentioned on line 184 requires a more detailed explanation to understand its specific function.

14. Cognition and Tau Laterality: On line 259, was there any observed difference in cognitive performance based on the lateralization of tau pathology (left vs. right hemisphere)?

Discussion:

- Tau-Connectivity Interpretation (Line 278): Exercise caution in interpreting the association between tau and connectivity. The current interpretation suggests that lower inter-hemispheric connectivity might drive tau propagation. However, it is also plausible that tau propagation itself leads to decreased inter-hemispheric connectivity. Ideally, assessing connectivity years before the onset of tau pathology would be necessary to establish the directionality of this relationship.

- Asymmetry Sample Size (Line 284): Are there any potential explanations for the observed difference in sample size between the right-asymmetric and left-asymmetric groups?

By addressing these points, the authors can further enhance the clarity, rigor, and interpretability of their valuable findings.

Reviewer #3

(Remarks to the Author)

Anijärvi et al. investigated whether asymmetric cerebral tau pathology is associated with (1) inter-hemispheric brain connectivity (functional: rsfMRI, and structural: diffusion MRI) or (2) asymmetric Abeta in A+ and T+ participants of the BIOFinder study.

Patients were categorized (according to temporal tau on tau PET) into left asymmetric (LA), symmetric (S), and right asymmetric groups (RA). The authors observed (1) no group difference in functional or structural connectivity, but (2) a strong association with asymmetry of Abeta (N = 233). In a longitudinal tau PET sample, baseline Abeta asymmetry predicted tau laterality over time.

The authors concluded that asymmetric tau deposition is not associated with weaker inter-hemispheric connectivity, but with asymmetric vulnerability to pathology.

The paper is very well written and deals with an important and, until now, open question regarding the asymmetric distribution of AD pathology seen frequently on clinical PET. The authors test well-defined hypotheses in an appropriate manner with strong data, i.e. large N with detailed characterization (APOE, cognition, etc.). A notable strength of this study is the validation of results in independent cohorts, which make the findings very convincing.

Comments and Questions

- Most of the findings are based on analyses that use tau laterality after conversion from a continuous to an ordinal variable (which is then used as a categorical variable for group comparisons). I have several questions related to this grouping:
 - Why was laterality of tau pathology not used as a continuous variable? This would avoid the need to exclude 5% of patients with a borderline LI, and it would also circumvent the inevitable loss of information when splitting a continuous variable. In the figures, the authors seem to agree, as the main finding (tau LI is associated with Abeta LI) is visualized as a scatter plot with continuous variables.
 - On page 5, line 107, it is stated that the grouping was based on 1 S.D. around LI=0. What was the mean LI (presumably not a perfect 0)?
 - On page 18, line 389: shouldn't it be 'LA if LI < Mean - LI' and 'RA if LI > Mean + 1 SD'?
- The vast majority of cases had symmetric tau deposition, already at this threshold. While the threshold of 1 S.D. seems appropriate, how many of the patients remain asymmetric when a threshold of 1.5 or 2 S.D. is applied?
- Related to this and regarding the choice of the statistical test: Can the authors please point out what the advantage of their selected statistical approach is (group comparison between symmetric and asymmetric tau groups, S vs LA and S vs RA)? Particularly in comparison to a partial Pearson correlation of the continuous variables (see the point above) with adjustment for baseline tau load.
- Might tau LI be best explained by a combination of connectivity and regional Abeta LI?
- Why did the authors include only patients who were 'not diagnosed as atypical AD' (page 17 line 369)? This might explain the high proportion of symmetric cases, and it might have diminished the sensitivity of the data analyses. Doesn't this mean that in the study there are no patients with asymmetric tau AND corresponding atypical clinical presentation? My speculation would be that the majority of cases with asymmetric tau deposition have an atypical clinical presentation. Can the authors please comment.
- Was the symptom duration available? The RA group has greater tau burden than the symmetric group — do they also have longer symptom duration? At which point of the individual symptom course were individuals scanned?
- In the longitudinal cohort, is asymmetry of Abeta or tau associated with faster increase of tau (not tau asymmetry) over time?
- The authors report that tau asymmetry increased over time. Do the authors see any hints for an inverse U-shaped function,

with increases, as reported, but decreases later in the disease course? (A further spread of tau over time might presumably lead to less pronounced asymmetry.)

- Overall, there are a lot of statistical tests in the manuscript. Can the authors please indicate which of the results survive correction for multiple tests? (I only found correction for tests between three groups, although only two of the possible pairs were tested, if I understood correctly: S vs RA and S vs LA)

- page 18, line 401, comparison of tau load between groups: Shouldn't this be an ANCOVA? Same page, line 404, Bonferroni-correction should be $0.05 / 3$, not $0.05 * 3$?

- Baseline tau LI predicted steeper cognitive decline (in Braak V-VI after adjustment for tau burden) — can the authors please suggest an interpretation of this finding? Can the authors please speculate on whether this effect might have been even greater if atypical AD were not excluded?

- Visualization of the data is overall excellent in my opinion, but I was surprised in Figures 5, 6, and 7 that regression lines have been plotted for cases with '-3 SD (= left asymmetry) and +3 SD (= right asymmetry)'. Do I understand correctly that LA and RA here have been defined differently from the rest of the manuscript (here: 3 SD, rest: 1 SD)? To plot regression lines only for extreme cases would seem somewhat unusual. Is it correct that the colorbar in Fig 5 indicates Abeta laterality as SD?

Version 1:

Reviewer comments:

Reviewer #1

(Remarks to the Author)

Thank you for the attention to my critique.

Reviewer #2

(Remarks to the Author)

Having reviewed the responses to my comments, I accept the revisions and recommend the article for publication.

Reviewer #3

(Remarks to the Author)

Thanks for answering my questions. My concerns have been perfectly addressed in the revisions.

made.

We sincerely thank the editor and reviewers for their constructive and insightful feedback on our manuscript. Their comments have significantly enhanced the quality of our work. We have carefully addressed all points raised and performed additional analyses as recommended. Below, we provide a detailed, point-to-point reply to each reviewer comment, with direct responses and clear indications of where changes have been made in the manuscript and supplementary materials. All significant edits are documented, and the relevant edited/added manuscript text has been “quoted and highlighted in yellow” for easy reference.

Reviewer #1

Comment 1

The results are strong in showing the amyloid-tau relationship. What's not as clear is how commonly this is seen. If I understand correctly, the BIOFINDER cohort began with 837 individuals who were amyloid+ which was then trimmed to 475 deemed to have elevated tau and 452 in whom asymmetry could be definitively addressed. Of these 146 were either left or right asymmetric; that is 17% of everyone with AD (defining AD as amyloid positive) or 31% of everyone with AD and elevated tau. This is an important group, but it's not the majority of people with AD. It is incumbent on the authors to point this out. It doesn't diminish the importance of the finding because it is a common one."

Response:

Thank you for this important comment. We apologize that we had not discussed this in the original submission. We have now addressed this in the Discussion, see below:

Added text in Discussion (page 17, lines 373-376):

"Importantly, asymmetric tau was observed in 32% (146/452) of A+T+ individuals, representing 17% (146/837) of the broader A β -positive cohort. While this subgroup does not constitute the majority of cases, its size (nearly one-third of those with elevated tau) underscores its clinical relevance."

Comment 2

Similarly the authors are not clear about how the samples were selected for the replication studies (OASIS, A4, ADNI). I may have missed it (could not find this information in the supplement), but please explain if there was any sort of similar restrictions on who was in the analysis for each cohort and the total study sample as a proportion of available participants with amyloid and tau PET data.

Response:

We appreciate this important comment. From all replication cohorts (OASIS, A4, ADNI), we include only participants with available baseline amyloid-PET and tau-PET scans and diagnostic classification. No additional inclusion/exclusion criteria were applied. Therefore, no changes in the manuscript were made.

Comment 3

Same issue for the longitudinal data depicted in figures 5 and 6. How were participants selected for inclusion and what proportion of participants with amyloid and tau data were included?

Response:

We thank again the reviewer for allowing us to clarify this. We now clarified that the 289 subjects included in the longitudinal analysis are a subgroup of the main cohort defined at the beginning of the Results and Methods sections. The only further inclusion/exclusion criteria were the availability of baseline A β -PET and more than one tau-PET scan.

Changes in Results (page 12, lines 253-256):

"To further investigate the impact of baseline A β laterality on longitudinal change in tau laterality, we examined a subsample of 289 A+ subjects from the original cohort of 837 who underwent at least one A β -PET scan and additionally had available longitudinal tau-PET scans (range of timepoints = 2-5; average follow-up = 2.9 years)."

Comment 4

The findings in figure 3 are very interesting. The authors did a nice job of showing that the strength of the amyloid/tau correlation is not the same across all regions. However an important remaining question is how specific are the regional associations between amyloid and tau? That is, in a given ROI (for example inferior temporal) how different is the correlation between the asymmetry of tau in the inferior temporal lobe and the asymmetry of amyloid in the inferior temporal lobe as opposed to (for example) tau in the inferior temporal lobe and amyloid asymmetry in the precuneus (or whole hemisphere for that matter)? It would not be difficult to run this analysis and depict it for all ROIs using a correlation matrix. I think many people would be interested in this result and it has consequences for mechanistic understanding of what's happening.

Response:

We thank the reviewer for the positive feedback and for the important comment. This is indeed a very interesting result to show to provide a better overview of the importance of specific regions and their A β -tau asymmetry associations across the brain. We included the most important results in the Results section; the figure itself is included in the supplementary materials in the interest of space. Overall, the new analysis showed that laterality in tau distribution, especially in temporal regions, is associated with laterality in A β distribution across the brain, although at different degrees.

Changes in Results (page 9-10, lines 205-214):

“Similar results were found when investigating the associations at a regional level with all regions (min $\beta=0.167$; mean $\beta=0.380$; max $\beta=0.633$; all $p_{FDR}<0.05$) except for two regions (the rostral anterior cingulate [$\beta=0.130$, $p_{FDR}=0.051$] and the hippocampus [$\beta=-0.034$, $p_{FDR}=0.608$]). The strongest effect sizes we found in the temporal regions, particularly the inferior temporal ($\beta=0.633$, $p_{FDR}<0.001$), fusiform ($\beta=0.613$, $p_{FDR}<0.001$), and middle temporal ($\beta=0.571$, $p_{FDR}<0.001$) gyri (Fig. 3b). Moreover, tau laterality appeared to be related to A β laterality not only in the same regions but across the brain. In particular, temporal regions showed the strongest average association with all brain regions (inferior temporal: $\beta_{avg}=0.428$; fusiform: $\beta_{avg}=0.405$; middle temporal: $\beta_{avg}=0.405$; see Fig. S2.8 in Supplementary S2).”

Added text and figure in Supplementary S2 (pages 16-17, lines 243-258):

“To examine the regional specificity of A β -tau asymmetry relationships, we performed a comprehensive pairwise region-by-region analysis between A β laterality and tau laterality (Fig. S2.8). This analysis addressed whether associations between A β and tau distributions are regionally specific (i.e., stronger within the same region) or reflect broader cross-regional relationships. Notably, the strongest same-region associations occurred in the temporal lobe, particularly the inferior temporal ($\beta=0.633$, $p_{FDR}<0.001$), fusiform ($\beta=0.613$, $p_{FDR}<0.001$), and middle temporal ($\beta=0.571$, $p_{FDR}<0.001$) gyri. Moreover, tau laterality in temporal regions showed the strongest associations with A β laterality across the entire brain. Specifically, tau laterality at inferior temporal gyrus demonstrated the highest average effect size with A β laterality across all regions ($\beta_{avg}=0.428$), followed by fusiform ($\beta_{avg}=0.405$), and middle temporal ($\beta_{avg}=0.405$) gyri.”

Figure S2.8. Region-by-region associations between A β laterality and tau laterality across the brain. All the coloured values indicate the effect sizes between regions that survived FDR-correction. Average effect sizes for all rows and columns are displayed at the edges of the matrix.

A β – amyloid-beta

Comment 5

I also wonder if the authors considered looking at a subgroup of nodes and edges in their connectivity analyses. Were the results the same if the authors confined their analyses to brain regions with the highest tau deposition or brain regions showing the greatest asymmetry?

Response:

This is an important point to address. To this purpose, we performed an additional analysis where we defined three new composite ROIs based on the bilateral tau burden within the A+T+ subjects (i.e., we examined a subgroup of nodes and edges based on

their group-average tau burden levels) – resulting in low, medium, and high tau burden ROIs. Similar to our original analyses, we did not find any significant associations between functional/structural connectivity and absolute tau laterality in none of the newly defined ROIs. This suggests that regardless of the region-specific vulnerability to tau deposition, the connectivity between hemispheres does not seem to be associated with an asymmetric distribution of tau. We added new text and figures to the manuscript and supplementary document to show this (see below).

Additionally, we stratified the nodes based on the group-average tau asymmetry values, resulting in low, medium, and high tau asymmetry ROIs. We found no significant associations between inter- or intra-hemispheric functional/structural connectivity and absolute tau laterality in neither of the ROIs. These new analyses were also added to the manuscript based on this important reviewer comment (see below).

Added text in Results (page 9, lines 186-196):

“Second, to address potential confounding effects of regional variability in tau load, three additional composite ROIs were defined based on the ranking of group-average tau SUVR values from A+T+ subjects (i.e., high tau burden corresponds to regions above the 75th percentile, medium tau burden corresponds to regions between the 25th and the 75th percentile and low tau burden corresponds to regions below the 25th percentile; see Supplementary S2). After adjusting for age, sex, and bilateral tau load, neither average inter-hemispheric (Fig. S2.5; functional: all $p_{\text{Bonf}} > 0.8$; structural: all $p_{\text{Bonf}} > 0.2$) nor intra-hemispheric connectivity (Fig. S2.6; functional: all $p_{\text{Bonf}} > 0.4$; structural: all $p_{\text{Bonf}} > 0.5$) showed significant associations with absolute tau laterality in any of these composite ROIs. Together, these sensitivity analyses confirm the null results we observed in the primary connectivity analyses.”

New text and figures in Supplementary S2 (pages 12-13, lines 181-225):

“Association between tau laterality and brain connectivity within meta-ROIs defined by tau burden severity

To test whether associations between tau laterality and average brain connectivity were affected by different levels of tau load in different regions, we conducted an additional analysis within three new composite ROIs. These ROIs were defined based on group-average bilateral tau load of the 452 A+T+ subjects. Regions were assigned to the low tau burden ROI if their group-average tau load was below the 25th percentile threshold (i.e., SUVR < 1.259), and to the high tau burden ROI if their tau load exceeded the 75th percentile threshold (i.e., SUVR > 1.637). Regions with values between these thresholds were assigned to the medium tau burden ROI. Next, we investigated average inter-hemispheric (Fig. S2.5) and intra-hemispheric (Fig. S2.6) functional and structural connectivity and their associations with the absolute tau laterality index using linear regression within each of these composite ROIs. All models were adjusted for age, sex,

and bilateral tau load, and p-values were Bonferroni-corrected for the number of composite ROIs (i.e., 3). Inter-hemispheric functional connectivity did not show any significant association with absolute tau laterality in any of the ROIs (Low tau: $\beta=0.006$, 95%CI=[-0.132; 0.144], $p_{\text{Bonf}}>0.900$; Medium tau: $\beta=0.068$, 95%CI=[-0.058; 0.195], $p_{\text{Bonf}}=0.860$; High tau: $\beta=0.057$, 95%CI=[-0.057; 0.170], $p_{\text{Bonf}}>0.900$); neither did structural connectivity (Low tau: $\beta=-0.110$, 95%CI=[-0.228; 0.009], $p_{\text{Bonf}}=0.212$; Medium tau: $\beta=-0.033$, 95%CI=[-0.144; 0.078], $p_{\text{Bonf}}>0.900$; High tau: $\beta=-0.003$, 95%CI=[-0.111; 0.105], $p_{\text{Bonf}}>0.900$). Similarly, there were no significant associations between absolute tau laterality and intra-hemispheric functional connectivity (Low tau: $\beta=0.010$, 95%CI=[-0.128; 0.148], $p_{\text{Bonf}}>0.900$; Medium tau: $\beta=0.081$, 95%CI=[-0.045; 0.207], $p_{\text{Bonf}}=0.618$; High tau: $\beta=0.082$, 95%CI=[-0.031; 0.195], $p_{\text{Bonf}}=0.468$) or structural connectivity (Low tau: $\beta=0.086$, 95%CI=[-0.037; 0.209], $p_{\text{Bonf}}=0.512$; Medium tau: $\beta=-0.017$, 95%CI=[-0.137; 0.104], $p_{\text{Bonf}}>0.900$; High tau: $\beta=-0.016$, 95%CI=[-0.115; 0.084], $p_{\text{Bonf}}>0.900$).

Figure S2.5. Association between absolute tau laterality index and inter-hemispheric functional/structural connectivity across three composite ROIs defined by group-average tau burden severity. Columns represent the three composite ROIs: low tau burden ($SUVR < 1.259$, 25th percentile), medium tau burden ($1.259 \leq SUVR \leq 1.637$), and high tau burden ($SUVR > 1.637$, 75th percentile). Regions included in each composite ROI, with

group-average tau and thresholds (dashed lines) for low/high tau classification. Middle row displays the analyses with functional connectivity and bottom row with structural connectivity. All models adjusted for age, sex, and bilateral tau load within the same composite ROI. Bonferroni-corrected p-values ($\times 3$) are annotated. Shaded bands indicate 95% confidence intervals. SUVR – standardised uptake value ratio

Figure S2.6. Association between absolute tau laterality index and intra-hemispheric functional/structural connectivity across three composite ROIs defined by group-average tau burden severity. Columns represent the three composite ROIs: low tau burden ($SUVR < 1.259$, 25th percentile), medium tau burden ($1.259 \leq SUVR \leq 1.637$), and high tau burden ($SUVR > 1.637$, 75th percentile). Regions included in each composite ROI, with group-average tau and thresholds (dashed lines) for low/high tau classification. Middle row displays the analyses with functional connectivity and bottom row with structural connectivity. All models adjusted for age, sex, and bilateral tau load within the same composite ROI. Bonferroni-corrected p-values ($\times 3$) are annotated. Shaded bands indicate 95% confidence intervals. SUVR – standardised uptake value ratio

Reviewer #2

Comment 1

Composite Region Definition and A β Threshold: For lines 378 and 412, please provide detailed information on the specific brain regions included in the composite measure. Additionally, clarify how the A β -PET neocortical uptake threshold of >1.033 was defined in the previous study. Specifically, was a two-Gaussian mixture model used? If so, what was the sample size used? Only controls were used to establish this threshold?

Response:

Thank you for the opportunity to provide greater detail on the PET methods. The threshold we used is indeed derived from a cut-off that was established on A β -PET data from the BioFINDER-2 cohort (including both cognitively unimpaired and impaired participants) using two-Gaussian mixture modelling as described in a previous study (see reference 52). We have now updated the relevant sentence to give the reader more complete information with additionally referring to the Supplementary Table S1.1 which now includes all the regions involved in that neocortical composite meta-ROI. Regarding the last question, the cut-off was not based only on controls, but cognitively unimpaired and impaired individuals.

Changes in Methods (page 20, lines 434-438):

“A β positivity was defined using a previously established cut-off (SUVR > 1.033) determined via Gaussian mixture modelling (GMM) applied on the A β -PET uptake in a neocortical composite meta-ROI (regions detailed in Table S1.1), based on data from both the cognitively unimpaired and impaired individuals in the BioFINDER-2 cohort [REF52]”

[REF52]: Brum WS, Cullen NC, Janelidze S, et al. A two-step workflow based on plasma p-tau217 to screen for amyloid β positivity with further confirmatory testing only in uncertain cases. *Nat Aging*. 2023;3(9):1079-1090. doi:10.1038/s43587-023-00471-5

Updated table in Supplementary S1 (page 2-3, line 38):

Table S1.1. Meta regions of interest.

Meta-ROI	Regions involved
Global	whole brain (i.e., all Desikan-Killiany regions)
Temporal	entorhinal cortex, parahippocampal cortex, fusiform cortex, amygdala, inferior temporal cortex, middle temporal cortex
Braak I-II	entorhinal cortex
Braak III-IV	parahippocampal cortex, fusiform cortex, amygdala, inferior temporal cortex, middle temporal cortex

Braak V-VI	caudal anterior cingulate cortex, caudal middle frontal cortex, cuneus, inferior parietal cortex, isthmus cingulate cortex, lateral occipital cortex, lateral orbitofrontal cortex, lingual cortex, medial orbitofrontal cortex, paracentral cortex, pars opercularis, pars triangularis, pars orbitalis, pericalcarine cortex, postcentral cortex, posterior cingulate cortex, precentral cortex, precuneus, rostral anterior cingulate cortex, rostral middle frontal cortex, superior frontal cortex, superior parietal cortex, superior temporal cortex, supramarginal cortex, frontal pole, temporal pole, transverse temporal cortex, insula
Early-Aβ	precuneus, posterior cingulate cortex, isthmus cingulate cortex, insula, medial orbitofrontal cortex, lateral orbitofrontal cortex
Intermediate-Aβ	banks of superior temporal sulcus, caudal middle frontal cortex, cuneus, frontal pole, fusiform cortex, inferior parietal cortex, inferior temporal cortex, lateral occipital cortex, middle temporal cortex, parahippocampal cortex, pars opercularis, pars orbitalis, pars triangularis, putamen, rostral anterior cingulate cortex, rostral middle frontal cortex, superior frontal cortex, superior parietal cortex, superior temporal cortex, supramarginal cortex
Late-Aβ	lingual cortex, pericalcarine cortex, paracentral cortex, precentral cortex, postcentral cortex
Neocortical composite	caudal anterior cingulate cortex, caudal middle frontal cortex, frontal pole, inferior parietal cortex, isthmus cingulate cortex, lateral orbitofrontal cortex, medial orbitofrontal cortex, middle temporal cortex, pars opercularis, pars orbitalis, pars triangularis, posterior cingulate cortex, precuneus, rostral anterior cingulate cortex, rostral middle frontal cortex, superior frontal cortex, superior parietal cortex, superior temporal cortex, supramarginal cortex

Comment 2

Temporal Meta-ROI Definition: Regarding lines 104 and 386, please provide more detailed information about the temporal meta-ROI. Specify the exact brain regions used to construct it and clearly define the cut-off criteria employed. Was this cut-off determined using the same tracer and data from the same study as the current analysis?

Response:

We thank the reviewer for pointing out that we need to add more detail on how we defined the tau positivity of the individuals within our study. We have now included more text in the Results section and refer to Table S1.1 in Supplementary S1 for the regions included in the temporal meta-ROI. Moreover, in the Methods section, we elaborated on how the tau positivity threshold was defined.

Changes in Results (page 5, lines 99-103):

“A cross-sectional sample of subjects with evidence of tau pathology (A+T+; n=475) based on unilateral (i.e., most affected hemisphere) tau-PET uptake in the temporal meta region of interest (meta-ROI; SUVR>1.362[REF27; REF28]; regions detailed in Table S1.1) was selected and categorized into three groups according to the spatial distribution of tau.”

Changes in Methods (pages 20-21, lines 445-450):

“The cross-sectional subsample of 475 participants included only A+ individuals who were also tau positive (T+) as defined by tau-PET uptake in a temporal meta-ROI (i.e., Braak stages I-IV regions; detailed in Table S1.1). The T+ threshold (SUVR > 1.362) was derived in the BioFINDER-2 cohort combining two approaches for cut-off estimation: 2 SD above the mean of A β -negative cognitively unimpaired older adults and GMM of the full cohort. [REF27; REF28]”

[REF27] Cho H, Choi JY, Hwang MS, et al. In vivo cortical spreading pattern of tau and amyloid in the Alzheimer disease spectrum. *Ann Neurol.* 2016;80(2):247-258.
doi:10.1002/ana.24711

[REF28] Leuzy A, Smith R, Ossenkoppele R, et al. Diagnostic Performance of RO948 F 18 Tau Positron Emission Tomography in the Differentiation of Alzheimer Disease From Other Neurodegenerative Disorders. *JAMA Neurol.* 2020;77(8):955-965.
doi:10.1001/jamaneurol.2020.0989

Comment 3

Lateralization Index (LI) Interpretation: On line 390, the statement that an absolute LI value <1 is considered symmetric is unclear. A symmetrical distribution should ideally have an LI of 0. Please clarify.

Response:

We thank the reviewer for this remark. We defined our tau asymmetry groups based on the group-average thresholds and indeed, the perfect symmetry would be $LI_{ref}=0$ (which we had used as the center point for our asymmetry threshold estimation). We decided to define individuals with tau LI less than 1 SD from either direction (i.e., left or right asymmetric tendency) as 1 SD is likely to represent a typical fluctuation in the measure and seemed to match the tau laterality spread of the tau negative individuals (see Fig. 1a). We have now clarified this point in the Methods section.

Changes in Methods (page 21, lines 452-456):

“Participants with LI exceeding ± 1 SD from perfect symmetry (i.e., $LI_{ref}=0$) were assigned into asymmetric groups – left tau asymmetric (LA) if $LI < LI_{ref} - 1$ SD, tau symmetric (S) if

$|LI| < LI_{ref} + 1 \text{ SD}$, or right tau asymmetric (RA) if $LI > LI_{ref} + 1 \text{ SD}$; subjects that had LI within the $\pm 5\%$ range of the threshold were dropped.”

Comment 4

Final Lateralization Threshold: Following the 5% range mentioned on line 391, what is the final threshold used to categorize asymmetry?

Response:

The final thresholds after considering the 5% borderline intervals were $LI > +9.70$ for right tau asymmetry, $LI < -9.70$ for left tau asymmetry, and $-8.78 < LI < +8.78$ for tau symmetry. The relevant sentence is in Methods (page 21, lines 443-444):

“The resulting threshold were: $|LI| > 9.70$ for asymmetry and $|LI| < 8.78$ for symmetry.”

Comment 5

Resting-State fMRI Preprocessing: The description of resting-state fMRI preprocessing on line 442 requires more detail. Please specify:

- *Whether slice-timing correction was applied.*
- *The method used for co-registration to standard space.*
- *The type of noise regression performed.*
- *Whether cerebrospinal fluid (CSF) and white matter signals were removed. If so, was mean signal regression or Principal Component Analysis (PCA) used, and if PCA, how many components were regressed out?*
- *Whether linear trends and motion components were accounted for.*
- *Whether spatial smoothing was applied, and if so, the kernel size used.*

Response:

Thank you for the opportunity to provide more detail on the fMRI analysis. Please find below the information requested:

- Slice-timing correction was applied using a routine included in AFNI
- Images were co-registered to MNI152 space via nonlinear registration using ANTs
- Nuisance regression included WM and CSF signals removed via mean regression (not PCA) along with 24 motion parameters (6 rigid-body + temporal derivatives) and linear trends
- Spatial smoothing was not applied

This information was incorporated in the Methods (page 23, lines 507-515):

“RSfMRI preprocessing utilised the Configurable Pipeline for the Analysis of Connectomes (C-PAC), with additional tools including AFNI and ANTs.[REF65; REF66; REF67] Processing included slice timing correction, T2-based unwarping for susceptibility distortion correction, and spatial normalisation to MNI152 standard space using nonlinear registration. Nuisance regression removed white matter (WM) and cerebrospinal fluid (CSF) signals via mean regression, along with 24 motion parameters (6 rigid-body + temporal derivatives) and linear trends. Data were bandpass filtered (0.01-0.1 Hz), and outlier frames were censored using a DVARS threshold (>75th percentile + 1.5×IQR). Participants exceeding mean/max frame-wise displacement thresholds (0.7/3.0 mm) were excluded. No spatial smoothing was applied.”

[REF65] Jon Clucas, Steve Giavasis, Amy Gutierrez, et al. FCP-INDI/C-PAC: C-PAC Version 1.3.0.post2 Beta. Published online May 14, 2024. doi:10.5281/ZENODO.593145

[REF66] Cox RW. AFNI: Software for Analysis and Visualization of Functional Magnetic Resonance Neuroimages. *Comput Biomed Res.* 1996;29(3):162-173. doi:10.1006/cbmr.1996.0014

[REF67] Tustison NJ, Cook PA, Holbrook AJ, et al. The ANTsX ecosystem for quantitative biological and medical imaging. *Sci Rep.* 2021;11(1):9068. doi:10.1038/s41598-021-87564-6

Comment 6

Inter-Hemispheric Connectivity Analysis: For lines 139 and 470, please elaborate on the implications of keeping only 10% of inter-hemispheric connections. If the weighted degree of these connections is computed, do the resulting connections cover all brain regions in the parcellation? Are there specific regions with a disproportionately higher weighted degree for inter-hemispheric connections compared to others? Including a supplementary figure illustrating the weighted degree of inter-hemispheric connections would be highly beneficial.

Response:

Thank you for highlighting the need for more detail on our brain connectome masking approach. The top 10% threshold was specifically chosen to preserve only the strongest connections (thereby reducing the likelihood of including connections that are not biologically real), while ensuring that no brain regions were segregated or excluded (i.e., all regions remain connected in the mask). We have now expanded Supplementary S1 with a new section that clearly explains the masking procedures, and we have added a figure illustrating the weighted degree of all regions in each of the four brain masks (see Fig. S1.1). Notably, lateral temporal regions display the highest weighted degrees in

functional connectivity, whereas medial posterior and frontal regions show the highest degrees in structural connectivity.

Changes in Methods (page 25, lines 540-543):

“To reduce noise and focus on the most biologically plausible connections, we masked both the FC and SC matrices for all subjects by selecting only the top 10% inter-hemispheric connections identified in 294 age-matched healthy controls with no evidence of A β or tau pathology from the BioFINDER-2 cohort (see Supplementary S1 for more information).”

New text and figure in the Supplementary S1 (pages 5-6, lines 78-95):

“Normative brain connectivity masks

For analyses investigating brain connectivity, normative brain masks were created from 294 age-matched, pathology-free healthy controls from the BioFINDER-2 cohort. These masks were generated by binarizing the averaged functional and structural connectomes to retain only the top 10% strongest connections, ensuring preservation of biologically meaningful pathways while effectively reducing noise from weaker, potentially spurious connections. The 10% threshold was chosen to be sufficiently strict to focus on the most robust connections, yet not so restrictive as to segregate or exclude any brain regions; all regions remain represented in the resulting masks. Whole-brain connectivity masks were first created for functional connectivity (Fig. S1.1a) and structural connectivity (Fig. S1.1b) analyses. Inter-hemispheric connectivity masks were subsequently defined by restricting to only between-hemisphere connections for functional connectivity (Fig. S1.1c) and structural connectivity (Fig. S1.1d).”

Figure S1.1. Weighted degree distribution of connectivity masks derived from healthy controls. Brain connectivity masks showing weighted degree (number of connections per region) for (a) whole-brain functional connectivity, (b) whole-brain structural connectivity, (c) inter-hemispheric functional connectivity, and (d) inter-hemispheric structural connectivity. All masks represent the top 10% strongest connections identified in 294 pathology-free BioFINDER-2 controls. The grey scale indicates weighted degree values.

Comment 7

Network-Based Statistic (NBS): On line 486, please clarify whether the NBS analysis was performed on a thresholded connectivity matrix. What the outcome is if the NBS analysis were applied to the unthresholded matrix.

Response:

Thank you for bringing this into our attention as we realise, we had not clarified that the NBS analysis was performed on the masked connectivity matrices. We have now updated the relevant sections in the manuscript and the supplementary document. Moreover, we have now also added a sensitivity analysis with NBS being performed on the unmasked connectivity matrices which brings us to the same conclusion - some minor within hemisphere connectivity differences emerged between the tau asymmetric and symmetric groups likely due to differences in tau load, but no inter-hemispheric differences were found.

Changes in Methods (page 26, lines 560-564 & 573-575):

“Network Based Statistic (NBS) method[REF82] was applied on FC and SC matrices to compare the whole-brain connectome differences between the tau asymmetry groups. The primary NBS analyses were performed on the masked matrices (retaining the top 10% strongest connections from healthy controls; see Supplementary S1 for more information on masking) to focus on biologically plausible connections.

[...]

Sensitivity analyses were additionally performed on the unthresholded connectivity matrices to ensure that the masking approach did not bias the results.”

[REF82] Zalesky A, Fornito A, Bullmore ET. Network-based statistic: Identifying differences in brain networks. *NeuroImage*. 2010;53(4):1197-1207.

doi:10.1016/j.neuroimage.2010.06.041

Added new text and edited the table and figure in Supplementary S2 (page 9, lines 138-143):

“Similar results were also found when the normative connectivity masks were not used (i.e., all connections were included in the analyses). Structural connectivity displayed more consistent difference between the right asymmetric and symmetric groups across the different statistical thresholds but, again, almost exclusively within the right hemisphere. This suggest that the lower structural connectivity in the right asymmetric group is an effect of the higher tau burden in that hemisphere (Fig. S2.3b).”

Table S2.1 and Figure S2.3 have been updated to also include the NBS results with the unmasked connectomes:

Table S2.1. NBS analysis comparisons of functional and structural connectivity between tau asymmetry groups. Each column represents a contrast between groups and each row shows the t-statistic threshold (t) used. Each cell represents the maximum detected component size C and the significance level i.e., C (p-value). A – tau asymmetric; LA – left tau asymmetric; S – tau symmetric – RA – right tau asymmetric; FC – functional connectivity; SC – structural connectivity.

Connectomes masked for top 10% whole-brain connections						
FC	A > S	A < S	LA > S	LA < S	RA > S	RA < S
t = 2.5	19 (p=0.045)	-	23 (p=0.041)	-	11 (p=0.072)	2 (p=0.239)
t = 3.0	2 (p=0.077)	-	9 (p=0.018)	-	3 (p=0.056)	1 (p=0.161)
t = 3.5	-	-	1 (p=0.053)	-	1 (p=0.054)	-
SC	A > S	A < S	LA > S	LA < S	RA > S	RA < S
t = 2.5	1 (p=0.684)	5 (p=0.104)	3 (p=0.231)	6 (p=0.074)	2 (p=0.423)	13 (p=0.031)
t = 3.0	-	2 (p=0.081)	1 (p=0.384)	1 (p=0.241)	2 (p=0.156)	4 (p=0.016)

t = 3.5	-	1 (p=0.073)	1 (p=0.127)	-	-	-
Sensitivity analysis: connectomes unmasked						
FC	A > S	A < S	LA > S	LA < S	RA > S	RA < S
t = 2.5	72 (p=0.077)	-	94 (p=0.050)	3 (p=0.489)	34 (p=0.158)	10 (p=0.295)
t = 3.0	11 (p=0.082)	-	26 (p=0.038)	-	12 (p=0.079)	1 (p=0.404)
t = 3.5	1 (p=0.155)	-	2 (p=0.076)	-	1 (p=0.167)	-
SC	A > S	A < S	LA > S	LA < S	RA > S	RA < S
t = 2.5	9 (p=0.452)	51 (p=0.062)	14 (p=0.426)	59 (p=0.038)	46 (p=0.255)	50 (p=0.038)
t = 3.0	1 (p=0.780)	7 (p=0.058)	1 (p=0.849)	5 (p=0.064)	7 (p=0.320)	15 (p=0.012)
t = 3.5	1 (p=0.360)	1 (p=0.165)	1 (p=0.457)	1 (p=0.119)	2 (p=0.364)	6 (p=0.002)

Figure S2.3. Components detected using NBS analysis between the tau asymmetry groups: (a) With connectomes masked to include only the top 10% whole-brain connections of the age-matched, pathology-free healthy controls (see Supplementary S1 for more information); (b) Without masking the connectomes (i.e., including all possible connections).

FC – functional connectivity; SC – structural connectivity; LA – left tau asymmetric; S – tau symmetric – RA – right tau asymmetric; C – component size; NBS – Network Based Statistic; t – threshold used in NBS

Comment 8

Threshold Consistency Across Studies: Regarding line 524, please confirm whether the different studies used to compute the A β positivity threshold employed the same approach (i.e., a two-Gaussian mixture model) as the previous data referenced in this study.

Response:

Based on this comment we have now updated the relevant sections with more details on the used cut-offs and correct references for the methods employed in the external cohort. For consistency with previous studies using data from OASIS-3, ADNI and A4, we opted to use the cut-offs predefined for each individual cohort.

Changes in Methods (page 28, lines 609-616):

“A β positivity was pre-defined in a cohort-specific manner for OASIS-3 and ADNI datasets. OASIS-3 used [11C]PiB (SUVR>1.42) and [18F]florbetapir (SUVR>1.19) tracers, as described previously.[REF83] ADNI used [18F]florbetaben (SUVR>1.08) and [18F]florbetapir (SUVR>1.11) tracers.[REF84] A4 used [18F]florbetapir PET for A β imaging and the ADNI-defined cut-off (SUVR>1.11) was applied to determine A β status. Tau imaging across all replication cohorts used [18F]flortaucipir PET. Tau positivity was defined as Temporal meta-ROI SUVR>1.34, a threshold established in prior work based on 2 SD from the mean tau-PET uptake within cognitively unimpaired A β -negative elderly in multiple combined cohorts.[REF85]”

[REF83] Su Y, Flores S, Wang G, et al. Comparison of Pittsburgh compound B and florbetapir in cross-sectional and longitudinal studies. *Alzheimers Dement Diagn Assess Dis Monit*. 2019;11(1):180-190. doi:10.1016/j.dadm.2018.12.008

[REF84] Schreiber S, Landau SM, Fero A, Schreiber F, Jagust WJ, for the Alzheimer’s Disease Neuroimaging Initiative. Comparison of Visual and Quantitative Florbetapir F 18 Positron Emission Tomography Analysis in Predicting Mild Cognitive Impairment Outcomes. *JAMA Neurol*. 2015;72(10):1183-1190. doi:10.1001/jamaneurol.2015.1633

[REF85] Ossenkoppele R, Rabinovici GD, Smith R, et al. Discriminative Accuracy of [18F]flortaucipir Positron Emission Tomography for Alzheimer Disease vs Other Neurodegenerative Disorders. *JAMA*. 2018;320(11):1151-1162. doi:10.1001/jama.2018.12917

Comment 9

Demographics Table: Is there a higher proportion of left-handed individuals in the right-asymmetric (RA) group compared to the left-asymmetric (LA) group?

Response:

That is a very interesting question, which we had not previously investigated. We have now added the information about handedness in the table for demographics (see Table 1). Statistically, there was no difference between the tau asymmetry groups in their handedness; numerically however, no left-handed were found in the right asymmetric group.

Updated Table 1 (pages 6-7, lines 129-137):

Table 1. Demographics.

Categorical variables have been presented as 'count (%)', normally distributed continuous variables as 'mean (SD)' and non-normally distributed variables as 'median [IQR]'. LA – left tau asymmetric; S – tau symmetric – RA – right tau asymmetric; M – male; F – female; R – right-handed; L – left-handed; A – ambidextrous; CU – cognitively unimpaired; MCI – mild cognitive impairment; AD – Alzheimer's disease; SUVR – standardised uptake value ratio; LI – laterality index; A β – amyloid-beta; MMSE – Mini-Mental State Examination; mPACC – modified Preclinical Alzheimer Cognitive Composite.
Missing data for 10^a, 42^b, 219^c, 1^d, and 42^e individuals.

		Cross-sectional A+T+ (n=452)			
		LA (n=102; 22%)	S (n=306; 68%)	RA (n=44; 10%)	P-value
Age, years		73.4 (6.7)	73.5 (7.3)	72.1 (6.9)	0.462
Sex	M	46 (45%)	135 (44%)	18 (41%)	0.895
	F	56 (55%)	171 (56%)	26 (59%)	
Education, years ^a		12.7 (3.82)	12.5 (4.0)	13.4 (4.1)	0.354
Handedness ^b	R	85 (94%)	262 (94%)	41 (100%)	0.567
	L	5 (6%)	16 (6%)		
	A		1 (<1%)		
Diagnosis	CU	11 (11%)	40 (13%)	9 (21%)	0.511
	MCI	39 (38%)	106 (35%)	12 (27%)	
	AD	52 (51%)	160 (52%)	23 (52%)	
Braak stage	I-II		11 (4%)		<0.001
	III-IV	21 (20%)	107 (35%)	6 (14%)	
	V-VI	81 (80%)	188 (61%)	38 (86%)	
Temporal tau, SUVR		1.9 [1.7, 2.3]	1.7 [1.4, 2.3]	2.1 [1.8, 2.7]	<0.001
Temporal tau LI		-14.4 (3.7)	-0.4 (4.9)	14.7 (5.4)	<0.001
Neocortical A β , SUVR ^c		1.59 (0.20)	1.63 (0.23)	1.56 (0.23)	0.211
APOE ϵ 4	0	32 (31%)	75 (25%)	16 (36%)	0.188
	1	49 (48%)	182 (59%)	21 (48%)	
	2	21 (21%)	49 (16%)	7 (16%)	
MMSE ^d		24.0 [21.0, 27.0]	25.0 [21.0, 27.0]	26.0 [22.7, 28.0]	0.328
mPACC ^e		-2.7 [-4.0, -1.7]	-2.6 [-3.8, -1.5]	-2.5 [-3.9, -1.2]	0.568

Comment 10

Tau and Inter-Hemispheric Connectivity: When comparing individuals with and without tau pathology, are there observable differences in inter-hemispheric connectivity? While the study investigates the association between connectivity changes and tau laterality, it is important to first establish whether tau accumulation itself alters these connections or only affects intra-hemispheric connections. If a control group without tau is not available, NBS correlations with continuous SUVR load shows any association between tau and inter-hemispheric connections?

Response:

We thank the reviewer for this insightful suggestion. To address whether tau pathology itself alters inter-hemispheric connectivity, we conducted new analyses comparing the cross-sectional A+T+ sample to 272 A-T- cognitively unimpaired controls from the BioFINDER-2 cohort (the same controls used to derive normative masks). After adjusting for age and sex, A+T+ individuals showed significantly lower whole-brain and inter-hemispheric functional/structural connectivity compared to controls. This confirms that tau burden is associated with widespread connectivity disruptions. These results are now detailed in Supplementary S2, strengthening the findings of our main analysis.

Added text in Results (pages 8-9, lines 176-182):

“While our primary analyses found no association between average brain connectivity and tau laterality, we further investigated whether connectivity relates to overall tau burden (see Supplementary S2). Compared to A-T- cognitively unimpaired controls, A+T+ individuals exhibited significantly lower average connectivity across the whole brain (functional: $\beta=-0.184$, $p=0.028$; structural: $\beta=-0.387$, $p<0.001$) including inter-hemispheric connections (functional: $\beta=-0.186$, $p=0.027$; structural: $\beta=-0.190$, $p=0.012$) after adjusting for age and sex (Fig. S2.4).”

Added text and figure in Supplementary S2 (pages 11-12, lines 161-180):

“Average connectivity differences between individuals with elevated tau burden and healthy controls

To evaluate general connectivity disruptions associated with elevated tau pathology, we compared functional and structural connectivity between the 452 A+T+ individuals and 272 A-T- cognitively unimpaired controls. Connectivity matrices were masked using normative whole-brain and inter-hemispheric masks (see Supplementary S1). Total whole-brain connectivity (averaged across all connections) was significantly reduced in A+T+ individuals (Fig. S2.4a) for both functional connectivity ($n=612$, $\beta=-0.184$, $95\%CI=[-0.349; -0.019]$, $p=0.028$) and structural connectivity ($n=646$, $\beta=-0.387$, $95\%CI=[-0.531; -0.243]$, $p<0.001$). Average inter-hemispheric connectivity was similarly

diminished in the A+T+ group (Fig. S2.4b) for functional connectivity ($\beta=-0.186$, 95%CI=[-0.351; -0.021], $p=0.027$) and structural connectivity ($\beta=-0.190$, 95%CI=[-0.338; -0.042], $p=0.012$). All models were adjusted for age and sex. These results demonstrate that tau pathology is associated with broad reduction in brain connectivity, providing critical context for investigating tau distribution related effects.”

Figure S2.4. Average functional and structural connectivity between the cross-sectional A+T+ sample and A-T- healthy controls: (a) Across the whole brain; (b) Within the inter-hemispheric connections. The connectomes were masked with the corresponding normative masks. A-T- – healthy pathology-free controls; A+T+ – individuals with evidence of amyloid-beta and tau pathology; * $p < 0.05$; *** – $p < 0.001$.

Comment 11

Contralateral Connections: On line 148, the reported number of 1412 connections between contralateral regions seems inconsistent with a parcellation of 42 regions, where one would expect a maximum of 42 homologous inter-hemispheric connections. Please clarify how these 1412 connections were defined. If the analysis did not specifically focus on these 42 homologous connections, exploring them might yield particularly interesting insights.

Response:

Thank you for identifying this inconsistency and for the constructive suggestion. In our original analysis, the term “contralateral connections” was incorrectly applied to all inter-hemispheric edges (including non-homologous regions) after masking. We acknowledge that this broad definition of “contralateral” was ambiguous and suboptimal, we have revised the analysis to focus specifically on homotopic connections (i.e., between homologous left-right regions, $n=36$ connections; the same regions that we investigated in the Ab-tau asymmetry analyses).

Considering also other reviewers' comments about the limiting aspects of comparing the tau asymmetry groups in the analyses instead of investigating the associations between continuous measures, we now updated this analysis which now focus on the A+T+ sample and examine the associations between the continuous measures.

We describe this updated analysis in the Results after the analyses of average inter-hemispheric connectivity by looking at homotopic inter-hemispheric connectivity region-by-region (i.e., edge-wise) and we did not find any significant associations between homotopic connectivity and absolute tau laterality after adjusting for age, sex, and bilateral tau load. After that, we similarly investigated whether there was a relationship between connectivity and bilateral tau load adjusted for age and sex to complement the previous question (see comment 10). We found significant associations between homotopic connectivity and bilateral tau burden which we described in the Sensitivity Analysis section of the Results.

Changed text in Methods (page 25, lines 553-558):

“Edge-wise associations between homotopic inter-hemispheric FC and SC (n=36 connections) and tau pathology were analysed using linear regressions. For each region, two models were tested to assess the relationships with (1) absolute tau laterality (OLS: homotopic FC/SC ~ age + sex + bilateral tau load + absolute tau LI) and (2) bilateral tau burden (OLS: homotopic FC/SC ~ age + sex + bilateral tau load). False Discovery Rate (FDR) correction (Benjamini-Hochberg method, $p < 0.05$) was applied across all 36 connections for both analyses.”

Changed text in Results (pages 7-8, lines 155-158):

“Notably, edge-wise analyses of homotopic (i.e., inter-hemispheric same-region) connectivity (n=36 connections) revealed no significant associations between absolute tau laterality and functional (all $p_{FDR} > 0.6$) or structural connectivity (all $p_{FDR} > 0.6$) (see Supplementary S2).”

Added text in Results in the sensitivity analysis section (page 9, lines 182-186):

“Edge-wise, within A+T+ individuals, lower functional and structural connectivity between homotopic regions was associated with higher bilateral tau burden in multiple regions across the brain, with the strongest effects observed in occipital and temporal areas for functional connectivity and across the neocortex except the temporal lobe for structural connectivity (Fig. S2.2).”

New text and figure in Supplementary S2 (pages 7-8, lines 110-129):

“Region-specific associations between connectivity and tau pathology

To assess region-specific relationships between inter-hemispheric brain connectivity and tau pathology, linear regressions were performed within the A+T+ group, examining associations of homotopic connectivity with both bilateral tau load and absolute tau

laterality. After adjusting for age and sex, higher tau burden was significantly associated with lower functional connectivity in occipital and temporal regions (Fig. S2.2), with the strongest effects observed in the superior parietal ($\beta=-0.299$, $p_{\text{FDR}}<0.001$), fusiform ($\beta=-0.287$, $p_{\text{FDR}}<0.001$), and lateral occipital ($\beta=-0.230$, $p_{\text{FDR}}<0.001$) gyri. For structural connectivity, lower connectivity was significantly related to higher tau burden in regions across the neocortex, particularly in the superior frontal ($\beta=-0.229$, $p_{\text{FDR}}<0.001$), superior parietal ($\beta=-0.218$, $p_{\text{FDR}}<0.001$), and precuneus ($\beta=-0.213$, $p_{\text{FDR}}<0.001$) gyri. Temporal regions did not show significant associations, except for a positive relationship in the superior temporal gyrus ($\beta=0.182$, $p_{\text{FDR}}=0.002$). In contrast, when examining associations between homotopic connectivity and absolute tau laterality (adjusted for age, sex, and bilateral tau load), no regions showed statistically significant relationships in functional connectivity (all $p_{\text{FDR}}>0.6$) or structural connectivity (all $p_{\text{FDR}}>0.6$).

Figure S2.2. Region-specific associations between homotopic connectivity and bilateral tau load adjusted for age and sex.
 FC – functional connectivity; SC – structural connectivity; FDR – false discovery rate.

Comment 12

A β -Tau Association in Temporal Regions: On line 178, the strongest effect sizes between tau and A β are reported in temporal regions. Do the individuals in this cohort exhibit tau pathology beyond these temporal regions? It would be helpful to include the number of individuals in each Braak stage in the demographics table. The strong correlation in temporal regions might be driven by the prevalence of individuals in earlier Braak stages. If individuals in Braak stages V-VI are specifically examined, is the effect size of the A β -tau association stronger within this temporal meta-ROI?

Response:

We agree that this is information about biological disease stage (i.e., Braak stage) is essential information that should be available to the reader. We now added the number of participants belonging to Braak I-II, III-IV, or V-VI stage to Table 1 (see comment 9, where the updated table can be found). Within our sample, most individuals are classified as Braak stage V-VI (81/102 left asymmetric, 188/306 symmetric and 38/44 right asymmetric). When we specifically examined the A β -tau asymmetry association within these individuals in Braak V-VI stages, we indeed see a slightly stronger effects both globally ($\beta=0.713$, $p<0.001$) and in temporal meta-ROI ($\beta=0.664$, $p<0.001$). Therefore, we believe our results are not biased by a high proportion of individual in earlier Braak stages.

Comment 13

SILA Algorithm Explanation: The SILA algorithm mentioned on line 184 requires a more detailed explanation to understand its specific function.

Response:

Thank you for highlighting the need for clarity regarding the SILA algorithm. We applied the Sampled Iterative Local Approximation (SILA) as an exploratory sensitivity analysis to investigate the hemispheric differences in A β pathology onset to see if this differs between individuals with asymmetric tau and subjects with symmetric tau. SILA models longitudinal A β -PET SUVR trajectories to estimate the timing of A β onset for each hemisphere, providing a statistical approximation rather than direct measurement (more information are provided in the Supplementary S2).

The analysis using the SILA model is now included in the Sensitivity Analyses subsection to emphasise their exploratory nature.

Changed text in Results (page 11, lines 241-248):

“To explore the estimated time differences in A β pathology onset between hemispheres in tau asymmetry groups, we applied the Sampled Iterative Local Approximation (SILA) algorithm on the full BioFINDER-2 cohort. SILA models individual A β accumulation curves to estimate the timing of hemispheric A β onset (see Supplementary S2). Asymmetric tau groups exhibited significantly larger inter-hemispheric differences in estimated global A β onset (left asymmetric: $\Delta=1.7$ years, $p_{\text{Bonf}}=0.006$; right asymmetric: $\Delta=2.5$ years, $p_{\text{Bonf}}<0.001$) compared to the symmetric group ($\Delta=1.3$ years; Fig. S2.13), suggesting earlier regional A β accumulation may contribute to tau lateralisation.”

Comment 14

Cognition and Tau Laterality: On line 259, was there any observed difference in cognitive performance based on the lateralization of tau pathology (left vs. right hemisphere)?

Response:

Thank you for this insightful suggestion. We have now added an analysis examining whether left- vs. right-predominant tau asymmetry differentially impacts cognitive decline to Supplementary S2 and referred to it in the main manuscript. Using non-absolute laterality indices (more positive = more right asymmetry, more negative = more left asymmetry; similarly to the longitudinal analysis between A β -tau asymmetries), we found no significant associations between the direction of tau laterality and mPACC scores. This confirms that the degree of tau asymmetry (magnitude), rather than its direction (left/right), underlies the observed relationship with cognitive decline.

Changed text in Results (page 14, lines 303-313):

“A+ participants showed distinct patterns of cognitive decline based on the degree of tau asymmetry (i.e., absolute laterality index). Higher baseline tau laterality was associated with steeper decline in modified Preclinical Alzheimer Cognitive Composite (mPACC) scores (Fig. 7a; n=259) in Braak III-IV ($\beta=-0.132$, 95%CI=[-0.171; -0.094], $p_{\text{Bonf}}<0.001$) and V-VI ($\beta=-0.157$, 95%CI=[-0.194; -0.121], $p_{\text{Bonf}}<0.001$). However, after adjusting for average tau uptake at the corresponding meta-ROIs, the independent effect of tau laterality on mPACC over time was statistically significant only in regions corresponding to Braak V-VI (Fig. 7b; $\beta=-0.104$, 95%CI=[-0.153; -0.055], $p_{\text{Bonf}}<0.001$). Direction of tau lateralisation (left vs right asymmetry) did not influence cognitive trajectories (see Fig. S2.15 in Supplementary S2). A β laterality did not have any significant effect on mPACC scores (Fig. 7c). See Supplementary Table S2.8 for an overview of the models.”

New text and figure in Supplementary S2 (page 30, lines 432-446):

“Directionality of tau and A β laterality and cognition

To assess whether left- or right-predominant pathological asymmetry differentially impacts cognition, we investigated the effect of baseline tau and A β laterality on mPACC scores using the non-absolute laterality indices (more positive values = more right asymmetry; more negative values = more left asymmetry). The direction of the tau lateralisation did not lead to different trajectories of mPACC scores in any regions (Fig. S2.15ab). Similarly, directionality of A β laterality had no effect on cognition (Fig. S2.15c).“

Figure S2.15. Longitudinal analysis within A+ sample over Braak meta-ROIs predicting mPACC score over time with: (a) Baseline tau laterality; (b) Baseline tau laterality after adjusting for tau load; (c) Baseline A β laterality after adjusting for tau laterality, tau load, and A β load.

The statistical annotations indicate the effect size and significance level of the interaction between time and baseline tau/A β laterality on cognitive assessment score. For visualisation, regression lines with 95% CIs of ± 2 SD baseline tau/A β laterality were plotted. A β – amyloid-beta; mPACC - modified Preclinical Alzheimer Cognitive Composite.

Comment 15

Tau-Connectivity Interpretation (Line 278): Exercise caution in interpreting the association between tau and connectivity. The current interpretation suggests that lower inter-hemispheric connectivity might drive tau propagation. However, it is also plausible that tau propagation itself leads to decreased inter-hemispheric connectivity. Ideally, assessing connectivity years before the onset of tau pathology would be necessary to establish the directionality of this relationship.

Response:

We thank the reviewer for this critical observation. We fully agree that our cross-sectional data cannot establish causality between connectivity and tau asymmetry – this was a poor choice of words from our part. To clarify this, we have revised the Discussion to emphasise that our null findings do not support a role for inter-hemispheric connectivity in explaining tau asymmetry but the full extent of the interaction between the lateralisation of tau pathology and connectivity cannot be investigated with our data.

Changed text in Discussion (page 15, lines 322-326):

“These null findings suggest that inter-hemispheric connectivity differences do not account for the asymmetric distribution of tau in individuals along the AD continuum. However, connectivity-based spreading of tau may still occur during the very early stages of the disease, before the pathology becomes detectable with PET.[REF26]”

[REF26] Meisl G, Hidari E, Allinson K, et al. In vivo rate-determining steps of tau seed accumulation in Alzheimer's disease. *Sci Adv.* 2021;7(44):eabh1448. doi:10.1126/sciadv.abh1448

Added sentence to the limitations section in Discussion (page 18, lines 400-404):

“Furthermore, tau accumulation and propagation likely lead to reduction in connectivity, which could have obscured the effect we were trying to investigate. Again, the full extent of the associations between connectivity and tau could be investigated only with an extensive longitudinal study following participants converting across the different stages of the AD continuum from A-T- to A+T+.”

Comment 16

Asymmetry Sample Size (Line 284): Are there any potential explanations for the observed difference in sample size between the right-asymmetric and left-asymmetric groups?

Response:

Thank you for highlighting this. We observed a greater proportion of left-asymmetric cases compared to right-asymmetric cases (e.g., 102 left vs. 44 right in our cohort), a pattern consistent with prior studies reporting left-hemispheric predominance in tau pathology. We have now mentioned the prevalence of the left/right asymmetry cases in the Discussion linking it to the previous literature. However, we cannot provide any explanation of this difference based on our data.

Changed text in Discussion (page 17, lines 376-379):

“Consistent with previous studies investigating tau asymmetry,[REF7; REF9; REF13; REF16; REF17; REF18] we found that left-predominant asymmetry (22%; 102/452) was more common than right-predominant asymmetry (10%; 44/452), and that asymmetric tau distribution was associated with worse cognitive decline over time.”

[REF7] Vogel JW, Young AL, Oxtoby NP, et al. Four distinct trajectories of tau deposition identified in Alzheimer's disease. *Nat Med.* 2021;27(5):871-881. doi:10.1038/s41591-021-01309-6

[REF9] Lu J, Zhang Z, Wu P, et al. The heterogeneity of asymmetric tau distribution is associated with an early age at onset and poor prognosis in Alzheimer's disease. *NeuroImage Clin.* 2023;38:103416. doi:10.1016/j.nicl.2023.103416

[REF13] Tremblay C, Serrano GE, Intorcchia AJ, et al. Hemispheric Asymmetry and Atypical Lobar Progression of Alzheimer-Type Tauopathy. *J Neuropathol Exp Neurol.* 2022;81(3):158-171. doi:10.1093/jnen/nlac008

[REF16] Ossenkuppele R, Schonhaut DR, Schöll M, et al. Tau PET patterns mirror clinical and neuroanatomical variability in Alzheimer's disease. *Brain*. 2016;139(5):1551-1567. doi:10.1093/brain/aww027

[REF17] Tetzloff KA, Graff-Radford J, Martin PR, et al. Regional Distribution, Asymmetry, and Clinical Correlates of Tau Uptake on [¹⁸F]AV-1451 PET in Atypical Alzheimer's Disease. *J Alzheimers Dis*. 2018;62(4):1713-1724. doi:10.3233/JAD-170740

[REF18] Martersteck A, Ayala I, Ohm DT, et al. Focal amyloid and asymmetric tau in an imaging-to-autopsy case of clinical primary progressive aphasia with Alzheimer disease neuropathology. *Acta Neuropathol Commun*. 2022;10(1):111. doi:10.1186/s40478-022-01412-w

Reviewer #3

Comment 1

Most of the findings are based on analyses that use tau laterality after conversion from a continuous to an ordinal variable (which is then used as a categorical variable for group comparisons). I have several questions related to this grouping:

- *Why was laterality of tau pathology not used as a continuous variable? This would avoid the need to exclude 5% of patients with a borderline LI, and it would also circumvent the inevitable loss of information when splitting a continuous variable. In the figures, the authors seem to agree, as the main finding (tau LI is associated with Abeta LI) is visualized as a scatter plot with continuous variables.*
- *On page 5, line 107, it is stated that the grouping was based on 1 S.D. around $LI=0$. What was the mean LI (presumably not a perfect 0)?*
- *On page 18, line 389: shouldn't it be 'LA if $LI < \text{Mean} - LI$ ' and 'RA if $LI > \text{Mean} + 1 \text{SD}$ '?*
- *The vast majority of cases had symmetric tau deposition, already at this threshold. While the threshold of 1 S.D. seems appropriate, how many of the patients remain asymmetric when a threshold of 1.5 or 2 S.D. is applied?*
- *Related to this and regarding the choice of the statistical test: Can the authors please point out what the advantage of their selected statistical approach is (group comparison between symmetric and asymmetric tau groups, S vs LA and S vs RA)? Particularly in comparison to a partial Pearson correlation of the continuous variables (see the point above) with adjustment for baseline tau load.*

Response:

Thank you for this comprehensive and constructive feedback on our approaches to group the subjects. Our current approaches definitely come with limitations as the reviewer rightly mentions. Therefore, we now elaborate more on the rationale for our methodological decisions and additionally provide in-depth sensitivity analyses demonstrating that various different approaches lead to highly similar results. The replies to all the reviewer's questions within this comment are numbered from 1-4 that answer different parts of it.

1. The main advantage of the grouping approach was that it allowed us to comprehensively characterise the differences between the groups (e.g., demographics, cognitive scores). However, we agree that it is indeed a limiting factor when the associations we are investigating consist of continuous variables. To eliminate the bias of grouping approaches, we added two extra panels to Figure 2 which previously displayed the comparison of average connectivity between the

groups, but now also include the equivalent analyses performed with the continuous variables (i.e., similarly as our Ab-tau asymmetry association analyses). These linear regressions describe the associations between absolute global tau laterality index and average global inter-hemispheric connectivity and fractional anisotropy at white matter tracts; the models have been adjusted for age, sex, and global bilateral tau load. Importantly, the results are highly compatible i.e. lead to the same conclusion as the analysis with the grouping approach. The relevant sections in the Results and Methods have also now been restructured for clarity (see below).

Changed text and figure in Results (pages 7-8, lines 147-158 & 163-174):

“No statistically significant differences were found in average inter-hemispheric functional connectivity (n=318; S-LA: $\beta=-0.151$, 95%CI=[-0.417; 0.116], $p_{\text{Bonf}}=0.802$; S-RA: $\beta=0.076$, 95%CI=[-0.312; 0.463], $p_{\text{Bonf}}>0.9$) or structural connectivity (n=352; S-LA: $\beta=0.135$, 95%CI=[-0.107; 0.376], $p_{\text{Bonf}}=0.823$; S-RA: $\beta=0.170$, 95%CI=[-0.162; 0.502], $p_{\text{Bonf}}>0.9$) between the tau asymmetric groups and the tau symmetric group (Fig. 2c). Similarly, no associations were found between absolute global tau laterality index and average inter-hemispheric functional connectivity ($\beta=0.090$, 95%CI=[-0.027; 0.207], $p=0.130$) or structural connectivity ($\beta=-0.037$, 95%CI=[-0.144, 0.071], $p=0.502$) when all A+T+ individuals were assessed (Fig. 2a). Notably, edge-wise analyses of homotopic (i.e., inter-hemispheric same-region) connectivity (n=36 connections) revealed no significant associations between absolute tau laterality and functional (all $p_{\text{FDR}}>0.6$) or structural connectivity (all $p_{\text{FDR}}>0.6$) (see Supplementary S2).”

[...]

“We further investigated microstructural integrity within the main white matter tracts connecting the two hemispheres but found no statistically significant associations between absolute global tau laterality and fractional anisotropy (Fig. 2b) - in the corpus callosum ($\beta=-0.047$, 95%CI=[-0.163, 0.069], $p=0.424$), forceps major ($\beta=0.022$, 95%CI=[-0.093, 0.137], $p=0.708$), or forceps minor ($\beta=-0.021$, 95%CI=[-0.137, 0.095], $p=0.721$). This result was consistent with the comparison of the same measure between the groups defined based on the tau laterality index (Fig. 2d) - in the corpus callosum (S-LA: $\beta=0.042$, 95%CI=[-0.216; 0.300], $p_{\text{Bonf}}>0.9$; S-RA: $\beta=-0.134$, 95%CI=[-0.518; 0.251], $p_{\text{Bonf}}>0.9$), forceps major (S-LA: $\beta=-0.033$, 95%CI=[-0.289; 0.223], $p_{\text{Bonf}}>0.9$; S-RA: $\beta=-0.220$, 95%CI=[-0.599; 0.159], $p_{\text{Bonf}}=0.762$), or forceps minor (S-LA: $\beta=0.001$, 95%CI=[-0.258; 0.260], $p_{\text{Bonf}}>0.9$; S-RA: $\beta=-0.136$, 95%CI=[-0.520; 0.248], $p_{\text{Bonf}}>0.9$); similar results were found for mean diffusivity (see Fig. S2.1 in Supplementary S2).”

Updated Figure 2:

Figure 2. Association between average connectivity between hemispheres and asymmetry in tau distribution: (a) Inter-hemispheric functional/structural connectivity vs absolute global tau laterality; (b) Fractional anisotropy in main white matter tracts vs absolute global tau laterality; (c) Inter-hemispheric functional/structural connectivity between tau asymmetry groups; (d) Fractional anisotropy in main white matter tracts between tau asymmetry groups.

Note – 16 subjects were dropped from the latter analyses after visual quality control of the tract segmentation, resulting in $n=336$. LA – left tau asymmetric; S – tau symmetric – RA – right tau asymmetric.

Changed text in Methods (page 25, lines 544-552):

“For each subject, we then calculated the global inter-hemispheric FC and SC by averaging the connections between all the left nodes connecting to the right (Fig. 8). The associations between global absolute tau laterality and these connectivity averages were then investigated in the cross-sectional A+T+ sample using OLS multiple linear regressions (OLS: FC/SC ~ age + sex + average global tau load + absolute global tau LI). Subsequently, these connectivity measures were compared between the tau asymmetry groups (i.e., LA vs S, RA vs S, and LA vs RA; three OLS models: FC/SC ~ age + sex + average global tau load + group) with the significance level ($p < 0.05$) Bonferroni-corrected for 3 comparisons. Parallel analyses assessed FA and MD in inter-hemispheric tracts using identical models.”

- Regarding the grouping approaches, the mean LI indeed was not 0, but -2.016 (i.e., slightly left asymmetric), which was the reason we decided to use the "theoretical

perfect symmetry" i.e. $LI_{ref}=0$ as our "mean". The reason for that was that we did not want to bias our results from the fact that our cohort had slightly more left tau asymmetric than right asymmetric cases. Also, thank you for pointing out the error where we had stated 'LA if $LI < 1$ SD', which had to be 'LA if $LI < LI_{ref} - 1$ SD'. We now have updated this section in the Methods.

Changed text in Methods (page 21, lines 452-456):

“Participants with LI exceeding ± 1 SD from perfect symmetry (i.e., $LI_{ref}=0$) were assigned into asymmetric groups – left tau asymmetric (LA) if $LI < LI_{ref} - 1$ SD, tau symmetric (S) if $|LI| < LI_{ref} + 1$ SD, or right tau asymmetric (RA) if $LI > LI_{ref} + 1$ SD; subjects that had LI within the $\pm 5\%$ range of the threshold were dropped.”

3. Regarding the usage of different SD levels for grouping the individuals to tau asymmetry groups, we decided to use 1 SD to have enough statistical power for group comparisons (otherwise the tau asymmetry groups would have been too small). If using 1.5 SD for the threshold ($|LI| > \pm 14.32$), left asymmetric group would have been $n=48$ (10.3%) and right asymmetric group $n=15$ (3.2%). If using 2 SD for the threshold ($|LI| > \pm 18.94$), left asymmetric group would have been $n=12$ (2.5%) and right asymmetric group $n=7$ (1.5%).
4. Finally, we appreciate the valid concern regarding the potential reduction in statistical power from excluding individuals in the 5% borderline laterality index interval. This exclusion was methodologically important to us to ensure clean group distinctions. However, to show that this step does not affect our primary findings, we performed two main cross-sectional analyses without dropping these individuals ($n=475$ i.e., extra 23 individuals). First, we investigated the association between absolute global tau laterality and global inter-hemispheric connectivity and white matter microstructural integrity (original analyses shown on Figure 2ab). Similarly to the results with the original study sample, we found no significant association between tau laterality and average inter-hemispheric functional ($\beta=0.084$, $p=0.155$) or structural connectivity ($\beta=-0.034$, $p=0.529$) or microstructural integrity at corpus callosum ($\beta=-0.036$, $p=0.537$), forceps major ($\beta=0.029$, $p=0.614$), or forceps minor ($\beta=-0.013$, $p=0.819$) tracts. Second, we assessed the association between global A β laterality and tau laterality (original analysis on Figure 3a) and found a similarly robust association between the two ($\beta=0.639$, $p<0.001$). Therefore, we are confident that this sample selection procedure (which only affected the cross-sectional analyses), did not affect the primary findings of this study suggesting that the core conclusions remain valid regardless of the choice.

Comment 2

Might tau LI be best explained by a combination of connectivity and regional Aβ LI?

Response:

Thank you for this insightful suggestion. To evaluate whether region-specific tau laterality is best explained by a combination of connectivity and Aβ asymmetry, we performed additional analyses comparing two OLS multiple linear regression models:

- base model: absolute tau laterality index (LI) ~ Aβ LI + age + sex
- extended model: base model + homotopic connectivity (FC or SC)

For each of the 36 regions, we compared models using F-tests on residual variance, with FDR correction. Across all regions, adding either functional or structural connectivity measures alongside Aβ laterality index did not improve model fit. These results align with our earlier findings: while Aβ asymmetry strongly correlates with tau lateralisation, inter-hemispheric connectivity seems to not affect this relationship. This new analysis was included in the Supplementary S2 and referred to in the Results.

Added text to Results (pages 13-14, lines 297-301):

“Finally, we evaluated whether region-specific tau laterality was influenced by inter-hemispheric connectivity alongside Aβ asymmetry. Neither functional nor structural connectivity improved model fits (all $p_{FDR} > 0.6$) or reached significance as predictors, indicating negligible explanatory power beyond Aβ asymmetry (see Supplementary S2).”

Added text to Supplementary S2 (page 28, lines 398-410):

“Combining connectivity and Aβ laterality to explain tau laterality

To evaluate whether region-specific tau laterality is best explained by a combination of connectivity and Aβ asymmetry, we performed additional analyses comparing two OLS multiple linear regression models: (1) base model: absolute tau laterality index (LI) ~ Aβ LI + age + sex; (2) extended model: base model + homotopic connectivity (FC or SC). For each of the 36 regions, we compared models using F-tests on residual variance, with FDR correction. Across all regions, adding functional connectivity did not improve model fit (ΔF : min=0.840, mean=0.996, max=1.329; all $p_{FDR} > 0.7$). Similarly, structural connectivity provided no significant improvement (ΔF : min=0.824, mean=1.006, max=1.138; all $p_{FDR} > 0.6$). Notably, connectivity measures themselves were never significant predictors in any of the models (all $p_{FDR} > 0.05$). These results align with our earlier findings: while Aβ asymmetry strongly correlates with tau lateralisation, inter-hemispheric connectivity seems not to affect this relationship.”

Comment 3

Why did the authors include only patients who were ‘not diagnosed as atypical AD’ (page 17 line 369)? This might explain the high proportion of symmetric cases, and it might have diminished the sensitivity of the data analyses. Doesn’t this mean that in the study there are no patients with asymmetric tau AND corresponding atypical clinical presentation? My speculation would be that the majority of cases with asymmetric tau deposition have an atypical clinical presentation. Can the authors please comment.

Response:

Thank you for pointing this out. Indeed, in the manuscript, the reference to the exclusion of atypical AD cases was confusing. The BioFINDER-2 cohort includes only a very limited number of atypical AD cases, but we did not explicitly exclude them.

In total, within our A+T+ sample (n=452), we had 7 individuals diagnosed with primary progressive aphasia (PPA) and 8 with posterior cortical atrophy (PCA). Notably, all individuals with right asymmetric tau distribution who were cognitively impaired (i.e., MCI or AD) were diagnosed with typical amnesic AD. However, within left tau asymmetric subjects, 4 were diagnosed with PPA and 1 with PCA. The group with symmetric tau deposition had 3 cases with PPA and 7 with PCA.

This information has now also been included in the Methods and Results sections in the manuscript.

Added text in Results (page 6, lines 124-128):

“The A+T+ sample included 15 cases with atypical presentation of AD. Among those with left-sided tau asymmetry, four individuals were diagnosed with PPA and one with PCA. Within the group characterized by symmetric tau deposition, three cases were diagnosed with PPA and seven with PCA. No atypical AD cases were identified in the right tau asymmetric group.”

Changed text in Methods (page 20, lines 426-430):

“The inclusion criteria for the present study were (1) evidence of A β pathology (A+); (2) age > 50 years; (3) did not fulfill the clinical criteria for other neurodegenerative diseases besides AD (e.g., frontotemporal dementia or Parkinson’s disease); (4) no other known severe neurological condition (e.g., brain tumour); (5) have at least one tau-PET scan.”

Comment 4

Was the symptom duration available? The RA group has greater tau burden than the symmetric group — do they also have longer symptom duration? At which point of the individual symptom course were individuals scanned?

Response:

Unfortunately, we do not collect data on symptom duration in the BioFINDER-2 cohort. However, as the aim of our study was to study AD pathophysiology, we have used tau burden as covariate in most of the statistical models to control for biological disease stage. Nevertheless, we agree that it would be beneficial to have information about symptom duration as a proxy of clinical disease severity. This limitation has now been included in the Discussion.

Added text in Discussion (page 18, lines 407-412):

“Finally, our cohort does not include symptom duration data, precluding analysis of whether group differences in tau burden reflect disease stage or are characteristic to tau lateralisation. However, we addressed this by adjusting for tau load in our statistical models, which serves as a proxy of clinical disease severity and helps distinguish stage-related effects from lateralisation-specific processes.”

Comment 5

In the longitudinal cohort, is asymmetry of Aβ or tau associated with faster increase of tau (not tau asymmetry) over time?

Response:

We thank the reviewer for this interesting question. We have now performed an extra analysis for addressing this point. The results revealed that higher baseline absolute tau laterality (but not Aβ laterality) is associated with faster longitudinal increases in bilateral tau burden. Specifically, individuals with higher tau asymmetry at baseline exhibited faster tau accumulation in Braak III-IV and Braak V-VI regions. In contrast, baseline Aβ laterality showed no significant association with tau burden progression. We have added a detailed description of these analyses to Supplementary S2.

New text in Results (page 13, lines 294-297):

“Additionally, we tested whether baseline tau or Aβ asymmetry predicted faster tau accumulation. Higher baseline absolute tau laterality was associated with a steeper increase in tau load in Braak III-IV ($\beta=0.085$, $p_{\text{Bonf}}<0.001$) and V-VI ($\beta=0.114$, $p_{\text{Bonf}}<0.001$) regions, while baseline Aβ laterality showed no effect (see Fig. S2.14).”

New text and figure in Supplementary S2 (page 29, lines 411-431):

“**Association between tau and Aβ laterality and longitudinal change in tau burden**
To assess whether asymmetry in tau or Aβ pathology predicts faster tau accumulation, we fitted two LME models in the longitudinal Aβ-positive cohort: for tau laterality (LME: $\text{tau load} \sim \text{time} * (\text{age}_{\text{baseline}} + \text{sex} + \text{A}\beta \text{ load}_{\text{baseline}} + \text{tau LI}_{\text{baseline}}) + [1 + \text{time} | \text{participant}]$)

and for A β laterality (LME: tau load ~ time * (age_{baseline} + sex + A β load_{baseline} + A β LI_{baseline}) + [1 + time | participant]), with Bonferroni correction applied to p-values for multiple comparisons across the meta-ROIs analysed. We found that higher baseline absolute tau laterality predicted faster tau burden increases in Braak III-IV ($\beta=0.085$, $p_{\text{Bonf}}<0.001$) and Braak V-VI ($\beta=0.114$, $p_{\text{Bonf}}<0.001$) regions but not in Braak I-II (Fig. S2.14a). In contrast, A β laterality did not display any significant effect on longitudinal change in tau burden in any of the meta-ROIs (all $p_{\text{Bonf}}>0.200$; Fig. S2.14b). These results indicate that tau lateralisation is linked to accelerated pathological disease progression in affected regions, a relationship independent of A β asymmetry.”

Figure S2.14. Longitudinal analysis of the association between pathological asymmetry and bilateral tau load across meta-ROIs within the A+ sample: (a) association between absolute tau laterality at baseline and tau load; (b) association between absolute A β laterality at baseline and tau load. The statistical annotations indicate the effect size and significance level of the interaction between time and baseline tau/A β laterality on bilateral tau load. For visualisation, regression lines with 95% CIs of baseline A β /tau laterality being 0 and 2 SD were plotted. A β – amyloid-beta; SUVR – standardised uptake value ratio.

Comment 6

The authors report that tau asymmetry increased over time. Do the authors see any hints for an inverse U-shaped function, with increases, as reported, but decreases later in the disease course? (A further spread of tau over time might presumably lead to less pronounced asymmetry.)

Response:

We thank the reviewer for raising this important question about potential nonlinear trajectories of tau asymmetry. To address this, we performed additional analyses comparing linear and quadratic mixed-effects models across the subsamples used in the longitudinal analyses. While we did not observe inverse U-shaped pattern (i.e., asymmetry increasing then decreasing), we identified accelerated tau lateralisation in individuals transitioning from A+T- to A+T+ status.

In specific, to assess temporal trajectories of absolute tau laterality, we compared linear (LME: $\tau \text{ LI} \sim \text{time} + [1 + \text{time} \mid \text{participant}]$) and quadratic (LME: $\tau \text{ LI} \sim \text{time} + I(\text{time}^2) + [1 + I(\text{time}^2) \mid \text{participant}]$) LME models across samples. Model fits were evaluated using Akaike Information Criterion (AIC). Effects of tau laterality over time were assessed with Bonferroni correction applied to p-values for multiple comparisons across the meta-ROIs compared (see the Additional Response Figure below). In the full A β -positive sample, global tau laterality displayed a linear increase over time ($\beta=0.043$, $p=0.002$). For A+T- individuals, tau laterality in Braak III-IV showed a linear increase in tau laterality ($\beta=0.114$, $p_{\text{Bonf}}<0.001$) whereas in Braak V-VI time displayed also nonlinear change over time ($\beta=0.043$, $p_{\text{Bonf}}=0.001$). A+T+ individuals displayed only linear longitudinal change in tau laterality at Braak V-VI ($\beta=0.061$, $p_{\text{Bonf}}=0.003$). Among individuals who remained A+T- throughout the study, only Braak III-IV exhibited minor linear tau laterality increases ($\beta=0.074$, $p_{\text{Bonf}}=0.045$). Notably, subjects who displayed progression in tau pathology (A+T- to A+T+) demonstrated linear increase of tau laterality over time in Braak III-IV ($\beta=0.250$, $p_{\text{Bonf}}<0.001$) but accelerating increase in Braak V-VI ($\beta=0.075$, $p_{\text{Bonf}}<0.001$). These results suggest that individuals transitioning from A+T- to A+T+ status experience pronounced, often nonlinear increases in tau asymmetry, whereas those who remain A+T- or were A+T+ already at baseline exhibit milder linear changes. This dichotomy highlights distinct trajectories of tau lateralisation tied to the presence and progression of neocortical tau pathology.

In the interest of space, we have not currently included this analysis in the main documents, but if the reviewer thinks they significantly contribute to the interpretation of the data, we suggest to add them to the supplementary materials.

Additional Response Figure. Longitudinal changes over time in absolute tau laterality at meta-ROIs within the study samples: (a) whole A+ sample at global meta-ROI (i.e., whole-brain for each hemisphere); (b) A+T- subsample at Braak meta-ROIs; (c) A+T+ subsample at Braak meta-ROIs; (d) A+T- subsample who stay A+T- throughout their follow-up; (e) A+T- subsample who progress to A+T+ during their follow-up. The statistical annotations indicate the effect size and significance level of the interaction between time and baseline A β laterality on tau laterality. For visualisation, regression lines with 95% CIs of mean tau laterality were plotted. LF – linear function; QF – quadratic function; AIC – Akaike Information Criterion

Comment 7

Overall, there are a lot of statistical tests in the manuscript. Can the authors please indicate which of the results survive correction for multiple tests? (I only found correction for tests between three groups, although only two of the possible pairs were tested, if I understood correctly: S vs RA and S vs LA)

Response:

We appreciate the feedback on the lack of details on our statistical approach. To clarify our approach: all group comparisons (symmetric vs left asymmetric, symmetric vs right asymmetric, and left vs. right asymmetric) were Bonferroni-corrected ($p_{Bonf} < 0.05$). Analyses across different meta-ROIs (e.g. Braak I-II, Braak III-IV, Braak V-VI) also applied

Bonferroni correction. For region-wise analyses (e.g., Desikan-Killiany atlas), we used False Discovery Rate (FDR) correction ($p_{FDR} < 0.05$). Longitudinal models were Bonferroni-corrected for the number of meta-ROIs tested. All corrected p-values are now explicitly labelled with subscripts (p_{Bonf} , p_{FDR}) in the Results and figures. The explanations in Methods have been updated as following. As an example, we have added Figure 3 with the updated statistical annotations in here.

Updated Figure 3:

Figure 3. Association between asymmetry in Aβ and tau distribution: (a) Aβ and tau laterality averaging the PET uptake over the whole hemisphere; (b) Regional-specific asymmetries in Aβ and tau. Aβ – amyloid-beta; * – $p_{FDR} < 0.05$; ** – $p_{FDR} < 0.01$; *** – $p_{FDR} < 0.001$.

Changed texts in Methods (page 21, lines 467-469; page 26, lines 579-582; page 27, lines 585-588 & 598-600):

“The significance level ($p < 0.05$) was Bonferroni-corrected for the number of group comparisons performed (3 comparisons: LA vs S, RA vs S, and LA vs RA).”

“The statistical analyses were performed using OLS multiple linear regressions (OLS: τ LI \sim age + sex + Aβ LI), with the significance level ($p < 0.05$) Bonferroni-corrected for the number of meta-ROIs the analysis was performed on or FDR-corrected for region-specific analyses across all regions.”

“The statistical analyses were performed using Linear Mixed Effects (LME) models with random intercepts and slopes for time and participants (LME: τ LI \sim time * (age_{baseline} + sex + Aβ LI_{baseline}) + [1 + time | participant]), with the significance level ($p < 0.05$) Bonferroni-corrected for the number of meta-ROIs the analysis was performed on.”

“The significance level ($p < 0.05$) for all models was Bonferroni-corrected for the number of meta-ROIs the analysis was performed on.”

Comment 8

*page 18, line 401, comparison of tau load between groups: Shouldn't this be an ANCOVA? Same page, line 404, Bonferroni-correction should be $0.05 / 3$, not $0.05 * 3$?*

Response:

We thank the reviewer for this comment. Our approach using ordinary least squares (OLS) regression with covariates (e.g., age, sex) and group indicators is statistically equivalent to ANCOVA in this context. ANCOVA is a special case of linear regression where categorical predictors (groups) and continuous covariates (e.g., age, sex) are combined. By including group as a categorical variable (via dummy coding) alongside covariates, our OLS models explicitly test group differences in tau load while adjusting for confounders, fulfilling the same purpose as ANCOVA. This approach is widely accepted and mathematically identical to ANCOVA for such analyses.

Regarding the Bonferroni correction, we acknowledge the confusion caused by our original phrasing. We did not clarify that we multiplied the uncorrected p-values with the number of comparisons performed, not the significance threshold. For example, in an analysis with three comparisons, a raw $p = 0.016$ becomes $p_{\text{Bonf}} = 0.016 * 3 = 0.048$, which is statistically significant because $p_{\text{Bonf}} < 0.05$.

Comment 9

Baseline tau LI predicted steeper cognitive decline (in Braak V-VI after adjustment for tau burden) — can the authors please suggest an interpretation of this finding? Can the authors please speculate on whether this effect might have been even greater if atypical AD were not excluded?

Response:

We thank the reviewer for this insightful remark, as we had not discussed this point in detail previously. Our finding that baseline tau laterality in Braak V-VI regions predicts steeper cognitive decline, even after adjusting for total tau burden. This suggests that asymmetric tau distribution in advanced disease stages may play an independent role in driving cognitive deterioration, even beyond the effect of overall tau load.

Regarding atypical AD, as we noted in response to your earlier question, we did not specifically exclude atypical AD cases from our analyses. However, due to the cohort composition and the relatively low prevalence of atypical AD cases, the number of

atypical cases in our cohort is limited. It is plausible that the relationship between tau asymmetry and cognitive decline is even more pronounced among atypical AD cases. This may be particularly the case for logopenic variant primary progressive aphasia (the language variant of AD) and corticobasal syndrome (the motor variant of AD), in which marked asymmetry has been well established. Unfortunately, the current sample is not suitable to address this point, but this is an important future research direction. We updated the relevant Discussion sections.

Changed text in Discussion (page 17, lines 376-385):

“Consistent with previous studies investigating tau asymmetry, we found that left-predominant asymmetry (22%; 102/452) was more common than right-predominant asymmetry (10%; 44/452), and that asymmetric tau distribution was associated with worse cognitive decline over time. This association was largely driven by individuals exhibiting both higher average tau load and greater tau asymmetry. Nevertheless, greater asymmetry in tau deposition in late Braak stage regions was associated with steeper cognitive decline, even after adjusting for tau burden. In contrast, A β laterality did not have any effect on cognition. Together, these findings suggest that tau lateralisation in brain regions mainly affected in advanced disease stages might be associated with faster cognitive decline.”

Changed text in Discussion (page 18, lines 404-407):

“A third limitation is the relatively small number of subjects with atypical AD and asymmetrical tau distribution. For example, we cannot rule out that a larger sample size could have allowed us to uncover more subtle differences in structural or functional connectivity.”

Comment 10

Visualization of the data is overall excellent in my opinion, but I was surprised in Figures 5, 6, and 7 that regression lines have been plotted for cases with ‘-3 SD (= left asymmetry) and +3 SD (= right asymmetry)’. Do I understand correctly that LA and RA here have been defined differently from the rest of the manuscript (here: 3 SD, rest: 1 SD)? To plot regression lines only for extreme cases would seem somewhat unusual. Is it correct that the colorbar in Fig 5 indicates Abeta laterality as SD?

Response:

We thank the reviewer for the positive feedback on our visualisations. We also agree that the choice of using 3 SD was likely suboptimal. To clarify: the group definitions (S, LA, RA) were based on a 1 SD threshold as detailed in the Methods and these groups were used in the initial cross-sectional analyses. However, in Figures 5-7, the regression

lines were plotted at ± 3 SD to visualise the continuous interaction effects between laterality indices and time across wide range. This approach is standard for illustrating continuous interaction effects in regression-based models (based on continuous values) and we do not redefine asymmetry groups. Separately, the colorbars in Figure 5 and 6 represent raw $A\beta$ laterality index values, not SD units. We have revised the figures to use ± 2 SD for regression lines and updated the captions to explicitly clarify the colorbar interpretation (see Figures 5-7 down below).

Updated Figure 5:

Figure 5. Longitudinal analysis of the association between baseline $A\beta$ laterality and changes over time in tau laterality: (a) Whole A+ sample at global meta-ROI (i.e., whole-brain for each hemisphere); (b) A+T- subsample at Braak meta-ROIs; (c) A+T+ subsample at Braak meta-ROIs.

The statistical annotations indicate the effect size and significance level of the interaction between time and baseline $A\beta$ laterality on tau laterality; **p-values Bonferroni-corrected for the number of meta-ROIs analysed for each subsample.** For visualisation, regression lines with 95% CIs of ± 2 SD baseline $A\beta$ laterality were plotted. **Colorbar indicates baseline $A\beta$ laterality index.** $A\beta$ – amyloid-beta

Updated Figure 6:

Figure 6. Longitudinal analysis of the association between baseline A β laterality and changes over time in tau laterality at Braak meta-ROIs, stratified by conversion to T+ status: (a) A+T- subsample who stay A+T- throughout their follow-up; (b) A+T- subsample who progress to A+T+ during their follow-up. The statistical annotations indicate the effect size and significance level of the interaction between time and baseline A β laterality on tau laterality; p-values Bonferroni-corrected for the number of meta-ROIs analysed for each subsample. For visualisation, regression lines with 95% CIs of ± 2 SD baseline A β laterality were plotted. Colorbar indicates baseline A β laterality index. A β – amyloid-beta

Updated Figure 7:

Figure 7. Longitudinal analysis within A+ sample over Braak meta-ROIs predicting mPACC score over time with: (a) **Absolute** baseline tau laterality; (b) **Absolute** baseline tau laterality after adjusting for tau load; (c) **Absolute** baseline A β laterality after adjusting for tau laterality, tau load, and A β load.

The statistical annotations indicate the effect size and significance level of the interaction between time and baseline A β /tau laterality on cognitive assessment score; **p-values Bonferroni-corrected for the number of meta-ROIs analysed for each model**. For visualisation, regression lines with 95% CIs of baseline A β /tau laterality being 0 and **2 SD** were plotted. A β – amyloid-beta; mPACC - modified Preclinical Alzheimer Cognitive Composite.